# Data Pruning Can Do More:
# A Comprehensive Data Pruning Approach for Object Re-identification

**Zi Yang**                                                           *zi.yang.1@campus.tu-berlin.de*
*Technical University of Berlin*
*Huawei Munich Research Center*

**Haojin Yang**                                                            *Haojin.Yang@hpi.de*
*Hasso Plattner Institute*

**Soumajit Majumder**                                          *soumajit.majumder@huawei.com*
*Huawei Munich Research Center*

**Jorge Cardoso**                                                  *Jorge.Cardoso@huawei.com*
*Huawei Munich Research Center*
*CISUC, University of Coimbra*

**Guillermo Gallego**                                           *guillermo.gallego@tu-berlin.de*
*Technical University of Berlin*
*Einstein Center Digital Future*
*Science of Intelligence Excellence Cluster*

**Reviewed on OpenReview:** *https://openreview.net/forum?id=vxxi7xzzn7*

## Abstract

Previous studies have demonstrated that not each sample in a dataset is of equal importance during training. Data pruning aims to remove less important or informative samples while still achieving comparable results as training on the original (untruncated) dataset, thereby reducing storage and training costs. However, the majority of data pruning methods are applied to image classification tasks. To our knowledge, this work is the first to explore the feasibility of these pruning methods applied to object re-identification (ReID) tasks, while also presenting a more comprehensive data pruning approach. By fully leveraging the logit history during training, our approach offers a more accurate and comprehensive metric for quantifying sample importance, as well as correcting mislabeled samples and recognizing outliers. Furthermore, our approach is highly efficient, reducing the cost of importance score estimation by 10 times compared to existing methods. Our approach is a plug-and-play, architecture-agnostic framework that can eliminate/reduce 35%, 30%, and 5% of samples/training time on the VeRi, MSMT17 and Market1501 datasets, respectively, with negligible loss in accuracy ($< 0.1\%$). The lists of important, mislabeled, and outlier samples from these ReID datasets are available at https://github.com/Zi-Y/data-pruning-reid.

## 1 Introduction

Object re-identification (ReID) is a computer vision task that aims to identify the same object across multiple images. ReID has a wide range of applications in security surveillance (Khan et al., 2021), robotics (Wengefeld et al., 2016), and human-computer interaction (Wang et al., 2019) among others. Similar to several other downstream computer vision applications, the performance of ReID algorithms is contingent upon the quality of data. ReID datasets are constructed by first collecting a pool of bounding boxes containing the

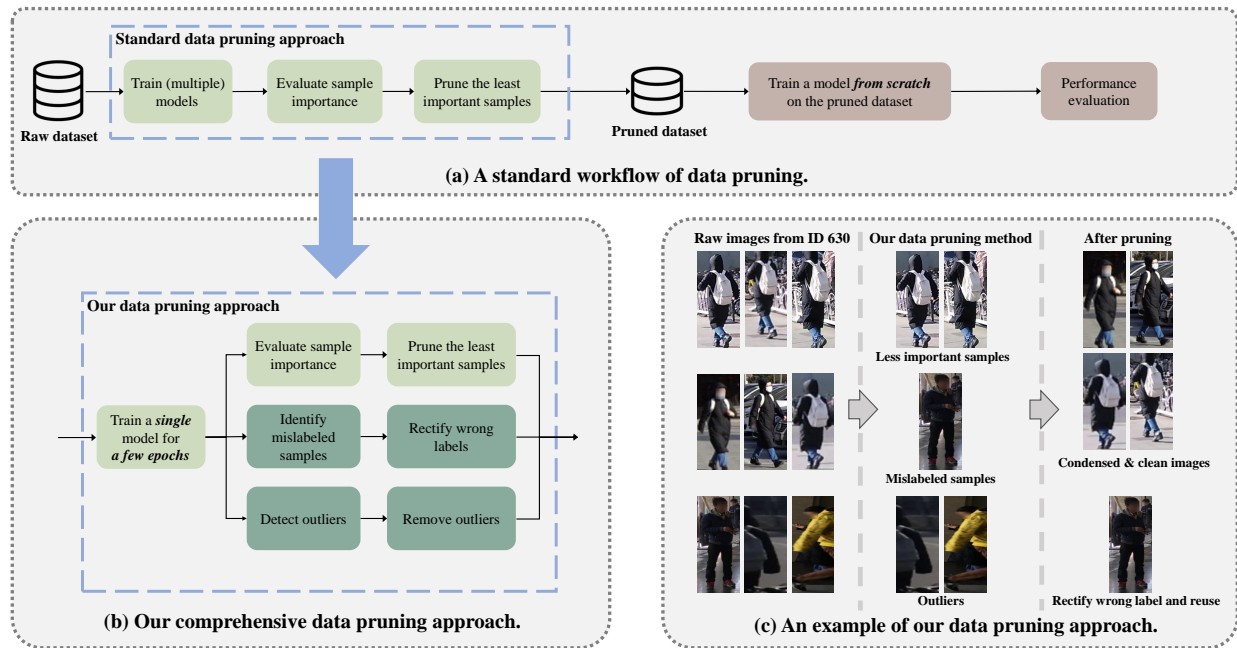

Figure 1: (a) The workflow of data pruning. (b) Our data pruning approach not only identifies *less important* samples, but also rectifies *mislabeled* samples and removes *outliers* (boxes highlighted in turquoise). (c) demonstrates an example of our method, where all images are from the same person (id 630, MSMT17 dataset). Our approach serves as a "pre-processing" step, reducing the dataset size to save storage and training costs of ReID models while having minimal impact on their accuracy.

intended object (e.g., people, vehicles); these images are typically obtained by applying category-specific object detectors to raw videos (Zheng et al., 2015; Fu et al., 2022). Then, the bounding boxes are assigned identity labels by manual annotators. ReID datasets created by this process generally present two issues:

1. **Less informative samples**: owing to the mostly fixed position of cameras, extracted bounding boxes from a sequence of images can contain redundant information, as these bounding boxes are likely to have the same background, lighting condition, and similar actions, e.g., Fig. 1(c). Such *less informative* samples from ReID datasets do not provide additional value to model training. Instead, they increase the training time leading to lowered training efficiency (Toneva et al., 2018; Feldman & Zhang, 2020; Paul et al., 2021; Sorscher et al., 2022).

2. **Noisy labels and outlier samples**: ReID datasets, typically containing a large number of people or vehicles with distinct appearances that require manual labeling, often suffer from noisy labels and outlier samples due to human annotation errors, ambiguities in images, and low-quality bounding boxes (Yu et al., 2019; Yuan et al., 2020). Such noisy samples and outliers can reduce inter-class separability in ReID models, thereby reducing their accuracy (Yu et al., 2019).

The issue of less informative samples is not limited to ReID datasets. Several methods (Toneva et al., 2018; Feldman & Zhang, 2020; Paul et al., 2021; Sorscher et al., 2022) have shown that a significant portion of training samples in standard image classification benchmarks (Krizhevsky et al., 2009; Deng et al., 2009) can be pruned without affecting test accuracy. To this end, various data pruning approaches (Sener & Savarese, 2017; Ducoffe & Precioso, 2018; Paul et al., 2021; Yang et al., 2022; Sorscher et al., 2022) have been proposed to remove unnecessary samples. A standard data pruning workflow is depicted in Fig. 1(a): 1) One or multiple models are trained on a raw dataset. 2) After training for some or all epochs, the importance score of each sample is estimated based on metrics, e.g., EL2N (Paul et al., 2021) - L2 norm of error, forgetting score (Toneva et al., 2018) - the number of times a sample has been forgotten during the training process. Generally, the more difficult the sample, the more important it is (Toneva et al., 2018;

Paul et al., 2021; Yang et al., 2022). 3) The samples with low importance scores are considered as easy or less informative samples and are discarded (Toneva et al., 2018; Feldman & Zhang, 2020; Paul et al., 2021). Steps 1 to 3 constitute a complete data pruning approach. 4) As a final step, we train the model *from scratch* on the pruned dataset and evaluate its performance. Despite these methods contributing significantly to finding less informative samples, they have been applied primarily to classification tasks. To the best of our knowledge, *no* existing studies have explored the application of these methods in ReID tasks, and their effectiveness in this new context still needs to be determined. Our work is dedicated to bridging the gap in applying (general) data pruning methods to ReID datasets.

Furthermore, the majority of these data pruning methods quantify the importance of samples at a specific training step (Paul et al., 2021; Yang et al., 2022; Sorscher et al., 2022) and tend to overlook the sample's behavior during the training, i.e., how the logits (un-normalized class probability scores) vary and evolve across epochs during training. We illustrate such a scenario in Fig. 2 and observe that the EL2N score (Paul et al., 2021) cannot differentiate between two *hard* samples, i.e., (b) and (c), as it relies solely on the model's prediction at the *last* training epoch, discarding the sample's training history. On the other hand, the forgetting score (Toneva et al., 2018) estimates the importance score by recording how many times the model *forgets* this sample, i.e., how many times a sample experiences a transition from being classified correctly to incorrectly during training. However, the forgetting score is still coarse-grained, as it exclusively focuses on tracking the number of forgetting events and does not leverage the full training information.

A further limitation of standard data pruning techniques is that they primarily emphasize the removal of easy samples and do *not* explicitly discard noisy samples or rectify mislabeled samples (Toneva et al., 2018; Paul et al., 2021; Sorscher et al., 2022). However, mislabeled samples and outliers are commonly found in ReID datasets (Yu et al., 2019; Yuan et al., 2020), which lead to performance degradation.

To address the limitations of these standard methods, we propose a comprehensive data pruning approach and apply it to ReID tasks. Our method not only prunes less informative samples more efficiently, but also corrects mislabeled samples and removes outliers in an end-to-end framework, as shown in Fig. 1(b). We hypothesize that instead of quantifying sample importance using discrete events (Toneva et al., 2018) or coarse inter-class relations (Paul et al., 2021; Sorscher et al., 2022) during training, fully leveraging the logit trajectory offers more insights. Accordingly, we propose a novel data pruning metric by fully utilizing the logit trajectory during training. Let us summarize our contributions:

1. To the best of our knowledge, we are the first to investigate the effectiveness of standard data pruning methods applied to ReID tasks in reducing dataset size while maintaining accuracy.

2. We propose a novel pruning metric that utilizes soft labels generated over the course of training to estimate robust and accurate importance score for each sample. Our method also demonstrates significant efficiency, reducing the cost of importance score estimation by a factor of ten compared to existing state-of-the-art methods.

3. Our proposed framework not only removes less informative samples but also corrects mislabeled samples and eliminates outliers.

4. We benchmark our comprehensive data pruning approach on three standard ReID datasets. It eliminates/reduces 35%, 30%, and 5% of samples/training time on VeRi, MSMT17 and Market1501 dataset, respectively, with negligible loss in accuracy ($< 0.1\%$).

## 2 Related Works

**Data Pruning** aims to remove superfluous samples in a dataset and achieve comparable results when trained on all samples (Toneva et al., 2018; Feldman & Zhang, 2020; Paul et al., 2021; Sorscher et al., 2022). Different to dynamic sample selection (Chen et al., 2017; Hermans et al., 2017) and online hard example mining (Shrivastava et al., 2016), which emphasize on informative sample selection for a mini-batch at each iteration, (static) data pruning aims to remove redundant or less informative samples from the dataset in one shot. Such methods can be placed in the broader context of identifying coresets, i.e., compact and informative data subsets that approximate the original dataset. $K$-Center (Sener & Savarese, 2017) and

supervised prototypes (Sorscher et al., 2022) employ geometry-based pruning methods, which estimate the importance of a sample by measuring its distance to other samples or class centers in the feature space. The forgetting score (Toneva et al., 2018) tracks the number of "forgetting events", as illustrated in Fig. 2 (c). GraNd and EL2N scores (Paul et al., 2021) propose to estimate the sample importance using gradient norm and error L2 norm with respect to the cross-entropy loss. The main limitations of these approaches are: (i) their importance estimations may be highly affected by randomness due to underestimation of the influence of the whole training history, leading to insufficient accuracy; and (ii) the significantly high computational overhead for the metric estimation required by these methods. For instance, the forgetting score (Toneva et al., 2018) and supervised prototype score estimation (Sorscher et al., 2022) require an additional number of training epochs equivalent to the whole training time.

**Object Re-identification (ReID)** aims at identifying specific object instances across different viewpoints based on their appearance (Zheng et al., 2016; Yadav & Vishwakarma, 2020; Zahra et al., 2022). ReID datasets (Zheng et al., 2015; Liu et al., 2016; Wei et al., 2018) consist of images of the object instances from multiple viewpoints typically extracted from image sequences. Therefore, images from the same sequence can often exhibit high similarity and contain redundant information i.e., the same background, consistent lighting conditions, and mostly indistinguishable poses. Furthermore, due to the subjective nature of manual annotation, ReID datasets often suffer from label noise and outliers (Yu et al., 2019; Yuan et al., 2020), such as heavily occluded objects and multi-target coexistence. During training, ReID tasks share similarity with the image classification task - given a dataset with a finite number of classes or identities, and models are trained to differentiate these distinct identities (Zheng et al., 2016; Yadav & Vishwakarma, 2020). As a result, the majority of ReID methods adopt the classification loss as an essential component (Wu et al., 2019; Si et al., 2019; Quispe & Pedrini, 2019; He et al., 2020). Owing to such similarity, several existing data pruning methods (Toneva et al., 2018; Paul et al., 2021; Sorscher et al., 2022) designed for image classification can directly be applied to ReID. As a first step, we apply existing pruning methods (Toneva et al., 2018; Paul et al., 2021) designed for the image classification task to ReID tasks. We observe that a portion of ReID datasets can be pruned without any noticeable loss in accuracy. Next, we propose a novel approach that offers improvements in terms of importance score estimation and computational overhead over existing methods. To the best of our knowledge, our work is the first to comprehensively study the efficiency of data pruning applied to ReID tasks.

## 3 Identifying Important Samples

### 3.1 Preliminaries

Our work focuses on supervised ReID. Similar to image classification (Zeng et al., 2014; Rawat & Wang, 2017), a standard ReID network includes a feature extraction backbone, e.g., CNN-based (He et al., 2016) or transformer-based model (Dosovitskiy et al., 2021), followed by a classification head (Song et al., 2019; Liao & Shao, 2022). The backbone network extracts representations from images of persons (Wei et al., 2018) or vehicles (Liu et al., 2016). The classification head, typically a fully connected layer, maps these extracted features to class labels, which in ReID refer to the individuals or vehicles in the respective dataset. The training of ReID models varies with respect to classification counterparts in the type of loss functions. In addition to the standard cross-entropy loss, ReID training often involves metric losses, e.g., triplet loss (Schroff et al., 2015) and center loss (MacDonald, 2013), to learn more discriminative features: $L_{\text{ReID}} = L_{\text{CE}} + \alpha L_{\text{metric}}$, where $\alpha$ is a hyper-parameter to balance the loss terms.

### 3.2 Motivation

Typically, data pruning approaches estimate the sample importance after training for several or all epochs. However, the majority of data pruning methods do not fully exploit or take into account the training dynamics in evaluating sample importance. We observe that underestimating the training dynamics can lead to less accurate and unreliable estimations of the sample's importance. Figure 2 illustrates the trajectories of logits for three samples: (a) a simple sample, (b) a hard sample, and (c) a 'harder' sample. Let us consider a classification task with three distinct classes, and each logit trajectory corresponds to one class. The difficulty

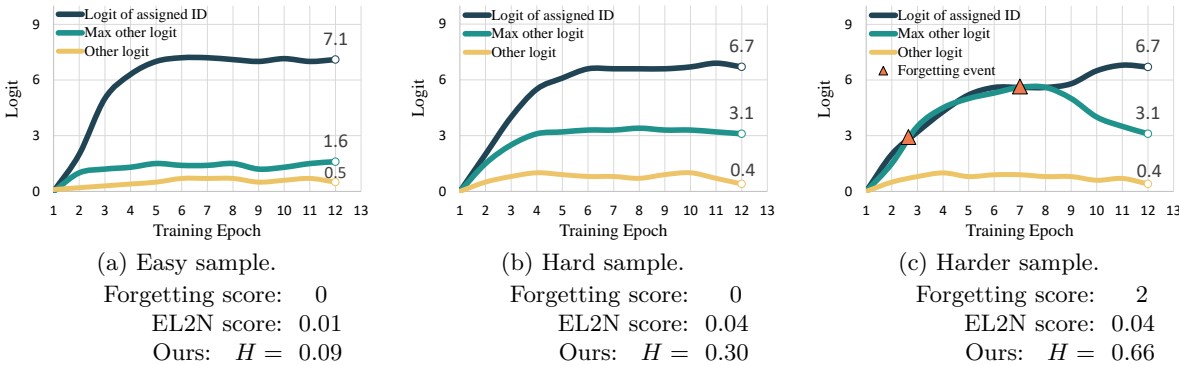

Figure 2: Logit trajectories for three samples (i.e., the evolution of the log probabilities of each sample belonging to each class over the course of training): (a) easy sample, (b) hard sample and (c) harder sample. There are three classes in total and each logit trajectory corresponds to one of such classes. The difference between the logits for each class can reflect the level of difficulty of this sample. In general, the more difficult the sample, the more important it is. The forgetting scores cannot distinguish (a) and (b), while the EL2N score relies solely on the model's prediction at the last epoch (thus without considering the history or "training dynamics"), hence it cannot differentiate between (b) and (c). Our approach fully exploits the training dynamics of a sample by utilizing the *average* logits values over all epochs to generate a more robust soft label. Then, the entropy of this soft label is employed to summarize the importance of the sample.

of a sample can be inferred by analyzing the discrepancy in logit values among different classes. Generally, the more difficult the sample, the more important it is (Toneva et al., 2018; Paul et al., 2021; Yang et al., 2022). EL2N (Paul et al., 2021) and other geometry-based methods (Sorscher et al., 2022; Yang et al., 2022) utilize the output of a trained model after a few or all training epochs to evaluate the importance of samples. However, a comprehensive importance evaluation cannot be achieved solely based on the model's output at a single epoch. As shown in Fig. 2 (b) and (c), although the last output logits of two hard samples are equal, the difficulties of learning them should not be equal: the model can correctly classify sample (b) from the early stage of training, while it struggles significantly and even mis-classifies sample (c) before the final training stage. Consequently, sample (c) in Fig. 2 should be considered more difficult or important. Inspired by this observation, we fully exploit the complete logit trajectories in order to achieve a more robust importance estimation, which we verify empirically. To be specific, for each sample, we leverage the complete logit trajectory to generate a soft label, which captures the relationship between a sample and all target classes. The entropy of the sample's soft label can represent the degree of confusion between different classes. A sample with high entropy could be wavering between different classes or a fuzzy boundary sample, thus making it more difficult or important.

### 3.3 Methodology

Let $(\mathbf{x}, y) \in \mathcal{D}_{\text{train}}$ be an image-label pair and $\mathbf{z}^{(t)}(\mathbf{x}) \in \mathbb{R}^c$ be the logit vector of dimension $c$ at epoch $t$, i.e., the un-normalized output of the classification head (pre-softmax). Here $c$ refers to the number of classes in the training set. A logit value, presented on a logarithmic scale, represents the network's predicted probability of a particular class (Anderson et al., 1988), with large values indicating high class probability. We first calculate the average logits of a sample over all training epochs $T$. We then apply Softmax $\sigma(\cdot)$ to generate the soft label $\tilde{\mathbf{y}}$ of this sample, which portrays the relationship between this sample and all classes,

$$\tilde{\mathbf{y}} = \sigma \left( \frac{1}{T} \sum_{t=1}^{T} \mathbf{z}^{(t)}(\mathbf{x}) \right) \in \mathbb{R}^c. \tag{1}$$

This soft label $\tilde{\mathbf{y}}$ summarizes the model's predictions at each training epoch. We conjecture that this soft label can effectively and accurately depict the ground truth characteristics of a sample. Next, we quantify

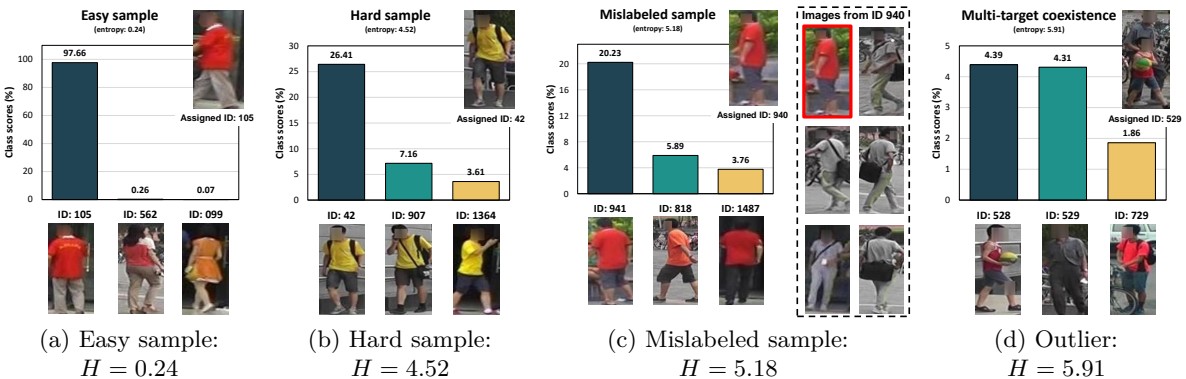

(a) Easy sample: $H = 0.24$     (b) Hard sample: $H = 4.52$     (c) Mislabeled sample: $H = 5.18$     (d) Outlier: $H = 5.91$

Figure 3: Illustration of the generated soft labels averaged over 12 training epochs for different sample types (a)–(d) and their entropies. Multi-target coexistence (d) is one such type of outlier. We show the top 3 identities. Notably, our soft label can accurately indicate the ground-truth label of the mislabeled sample (c) without being influenced by the erroneous label. Images are from Market1501 dataset.

the importance score of a sample using the soft label entropy,

$$H(\tilde{\mathbf{y}}) = -\sum_{i=1}^{c} p_i \log_2 p_i, \tag{2}$$

where $\tilde{\mathbf{y}} = (p_1, \ldots, p_c)^\top$ and $c$ is the total number of classes. The importance of a sample can generally be evaluated by the difficulty of this sample (Sener & Savarese, 2017; Toneva et al., 2018; Paul et al., 2021). The entropy is a straightforward metric to quantify the difficulty of a sample: A sample with a small entropy value implies that it contains the typical characteristics of a single class, and the classifier can easily distinguish this sample. In contrast, samples with larger entropy values could have different characteristics of more than one class at the same time (e.g., samples near classification boundaries), thus making it more difficult or important. Figure 3 illustrates our generated soft labels and the corresponding entropy of several typical samples. Please refer to Appendix E for more examples.

The Area Under Margin (AUM) (Pleiss et al., 2020) also leverages training dynamics, but for the identification of *noisy* samples in a dataset. The AUM is defined as the difference between the logit values for a sample's assigned class and its highest non-assigned class and averaged over several epochs, i.e., $\text{AUM}(\mathbf{x}, y) = \frac{1}{T} \sum_{t=1}^{T} \left( \mathbf{z}_y^{(t)}(\mathbf{x}) - \max_{i \neq y} \mathbf{z}_i^{(t)}(\mathbf{x}) \right)$, where $\mathbf{z}_i^{(t)}(\mathbf{x})$ corresponds to logit value of class $i$ at epoch $t$. This scalar metric primarily captures the knowledge related to the assigned class while ignoring the information across non-assigned classes. However, knowledge from non-assigned classes can potentially be of greater importance than that from the assigned class, as demonstrated in (Zhao et al., 2022). Different from AUM, our method utilizes information from all classes to depict a comprehensive relationship between a sample and the label space. Additionally, as the AUM metric does not fully encapsulate the characteristics of the sample, it is only capable of identifying mislabeled samples, but it cannot correct them. In the next section, we elaborate further on our approach for label correction and outlier detection.

## 4 Data Purification

Although most samples with high importance scores contribute significantly to model training, some high-scoring samples may be noisy (Feldman & Zhang, 2020; Paul et al., 2021) and degrade model performance. To address this limitation, in this section we further exploit the generated soft label to "purify" samples in two ways: by correcting mislabeled samples and by eliminating outliers.

**Dataset Noise.** Noise in ReID datasets can be broadly categorized into (Yu et al., 2019): (i) label noise (i.e., mislabeled samples) caused by human annotator errors, and (ii) data outliers caused by object detector errors (e.g., heavy occlusion, multi-target coexistence, and object truncation; see examples in Appendix C). Our method aims to reuse mislabeled samples by label correction and eliminating outliers.

**Correcting Mislabeled Samples.** We hypothesize that the soft label generated from the average logits of a sample can reflect the ground-truth characteristics of this sample accurately. Accordingly, we directly utilize soft label $\tilde{\mathbf{y}}$ from Eq. 1 to identify whether a sample is mislabeled or not. When the assigned label differs from the class with the highest score in the soft label, i.e., $\mathrm{argmax}(\tilde{\mathbf{y}}) \neq y$, the sample is considered as mislabeled. The ground-truth label of this sample is then accordingly corrected to $\mathrm{argmax}(\tilde{\mathbf{y}})$. Indeed, our ablation experiments offer empirical evidence that accumulated logits over epochs lead to a much more accurate prediction of the ground-truth label than solely relying on logits from a single epoch (more details in Sec. 5.6).

**Identifying Outliers.** The entropy of the soft label reflects the difficulty of a sample, allowing us to infer whether it is an outlier or not. A straightforward approach is to set a threshold on the entropy: if the entropy of the sample surpasses the predefined threshold, the sample is likely an outlier. However, theoretically, such an entropy threshold should be changed depending on the number of classes: the upper bound of the entropy is determined by the number of classes, i.e., $H(\tilde{\mathbf{y}}) \leq \log_2(c)$, rendering it unsuitable for use across datasets with different numbers of classes. Therefore, we employ the highest class score as an indicator to determine outliers. If the highest class score of a soft label is relatively low, i.e., $\max(\tilde{\mathbf{y}}) \leq \delta$, it is highly likely that the sample is an outlier. We conduct a sensitivity analysis to evaluate the impact of the threshold $\delta$ in Sec. 5.6 and observe our method is robust to this choice.

## 5 Experiments

We demonstrate the effectiveness of our method via detailed experimentation. In Sec. 5.3, we evaluate our pruning metric on ReID and classification datasets, and verify that it effectively quantifies the sample importance. In Sec. 5.4, we train models on noisy datasets after correcting labels and removing outliers to validate the data purification ability. In Sec. 5.5, we present the results of our comprehensive data pruning method, which includes removing less important samples, correcting mislabeled samples, and eliminating outliers. Finally, we perform several detailed ablation studies in Sec. 5.6 to evaluate the impact of hyper-parameters and verify the importance of logit accumulation.

### 5.1 Datasets and Evaluation

**Datasets.** We evaluate our method across three standard ReID benchmarks: Market1501 (Zheng et al., 2015) and MSMT17 (Wei et al., 2018) for pedestrian ReID, and VeRi-776 (Liu et al., 2016) for vehicle ReID. Market1501 is a popular ReID dataset consisting of $32,668$ images featuring $1,501$ pedestrians, captured across six cameras. Compared to Market1501, MSMT17 is a more challenging and large-scale person ReID dataset (Wei et al., 2018), comprising $126,441$ images from $4,101$ pedestrians. VeRi-776 is a widely used vehicle ReID benchmark with a diverse range of viewpoints for each vehicle; it contains $49,357$ images of 776 vehicles from 20 cameras.

**Evaluation.** Given a query image and a set of gallery images, the ReID model is required to retrieve a ranked list of gallery images that best matches the query image. Based on this ranked list, we evaluate ReID models using the following metrics: the cumulative matching characteristics (CMC) at rank-1 and mean average precision (mAP), which are the most commonly used evaluation metrics for ReID tasks (Bedagkar-Gala & Shah, 2014; Ye et al., 2021; Zheng et al., 2016). As an evaluation metric, we adopt the mean of rank1 accuracy and mAP (He et al., 2020), i.e., $\frac{1}{2}(\mathrm{rank1} + \mathrm{mAP})$, to present the results in a single plot.

### 5.2 Implementation Details

**Estimating the Sample Importance Score.** For ReID tasks, we follow the training procedure in (Luo et al., 2019): a ResNet50 (He et al., 2016) pretrained on ImageNet (Deng et al., 2009) is used as the backbone and trained on a ReID dataset for 12 epochs to estimate sample importance. For optimization, we use Adam (Kingma & Ba, 2014) with a learning rate of $3.5{\times}10^{-4}$. We do not apply any warm-up strategy or weight decay. The batch size is set to 64. Following Luo et al. (2019), We use a combination of the cross-entropy loss and the triplet loss as the objective function. During training, we record the logit values of each sample after each forward pass. Then, we generate the soft label of each sample based on Eq. 1 and

Table 1: Extra training epochs and time needed for metric estimation on a ReID dataset. The total training epochs for a ReID model is 120.

| Method | EL2N(20 models) | Forgetting score | Supervised prototypes | Ours |
|---|---|---|---|---|
| Extra training epochs | 240 | 120 | 120 | **12** |
| Extra training time[1] | 5.2 hours | 2.6 hours | 2.6 hours | **15.8 mins** |

calculate its entropy as the importance score. For classification tasks, we use the model architecture and training parameters from Toneva et al. (2018) for experiments on CIFAR100 dataset and from Boudiaf et al. (2020) on CUB-200-2011 dataset. In contrast to the ReID tasks, we do *not* employ any pre-trained models in classification tasks, i.e., the model weights are randomly initialized. Please refer to Appendix A and B for more implementation details.

**Data Pruning.** Given the importance scores of all samples, we sort them to obtain a sample ordering from easy (low importance score) to difficult (high importance score). Next, given the pruning rate, we remove the corresponding number of easy samples from the original training set. For instance, if the pruning rate is 10%, we remove the top 10% of the easiest samples. Subsequently, following Luo et al. (2019), we train four ReID models from scratch on this pruned dataset independently, each with a different random seed, and report their mean accuracy. The total number of training iterations is linearly proportional to the number of samples. Therefore, a reduction in sample size leads to a proportionate decrease in training time.

## 5.3   Find Important Samples

This section aims to verify the effectiveness of our proposed importance score by data pruning experiments. Because our work predominantly concentrates on data pruning applied to the ReID tasks, we demonstrate the efficacy of our proposed importance score on three ReID datasets in Sec. 5.3.1. Considering the substantial similarity between ReID and classification tasks, we also extend our data pruning experiments to CIFAR-100 and CUB-200-201 datasets in Sec. 5.3.2, thereby ensuring a more comprehensive validation of our method.

### 5.3.1   Data Pruning on ReID Datasets

We observe that the majority of data pruning methods have been evaluated on standard image classification datasets. However, to date, no pruning metric has been evaluated on ReID datasets. To this end, we perform a systematic evaluation of three standard pruning metrics on ReID datasets: EL2N score (Paul et al., 2021), forgetting score (Toneva et al., 2018), supervised prototypes (Sorscher et al., 2022). Following the standard protocol (Toneva et al., 2018; Sorscher et al., 2022), the forgetting scores and supervised prototype scores are calculated at the end of training. The EL2N score (Paul et al., 2021) and our proposed metric are estimated early in training (i.e., after 12 epochs - which is 10% of the total training epochs and recommended by the standard EL2N protocol). For EL2N, we report two scores: one generated by a single trained model and the other one by 20 trained models, each trained with a different random seed. Training epochs for the importance score estimation of standard data pruning methods are shown in Tab. 1. In addition to standard data pruning methods, we also conducted additional comparisons with RoCL (Zhou et al., 2020), which is a curriculum learning method designed for learning with noisy labels but also utilizes training dynamics to select samples. To fairly compare with it, we utilize its core idea of moving average loss as the pruning metric to realize static data pruning. The training sets are constructed by pruning different fractions of the lowest-scored samples. In all experiments, training on the full dataset (*no pruning*) and a random subset of the corresponding size (*random pruning*) are used as baselines (Sorscher et al., 2022). We present the results of data pruning approaches on three different ReID datasets in Fig. 4.

**Better Pruning Metric by Leveraging the Training Dynamics.**   The results in Fig. 4 validate our assumption that fully exploiting training dynamics can lead to a more precise and comprehensive evaluation of the sample importance. Our method surpasses all competing methods across all datasets. The importance

---

[1]Extra training time is measured using a single NVIDIA V100 GPU on MSMT17 dataset.

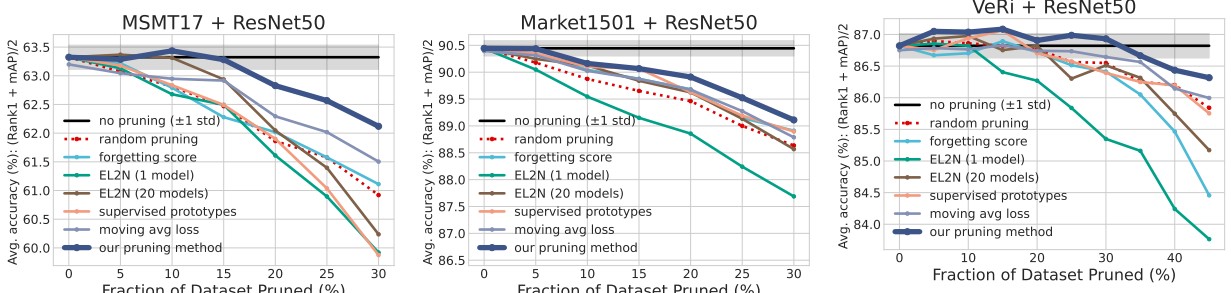

Figure 4: Data pruning on ReID datasets. We report the mean of Rank1 and mAP on 3 ReID datasets (labeled at the top), obtained by training on the pruned datasets. For each method, we carry out four independent runs with different random seeds and report the mean.

of incorporating the training dynamics is also observed in the forgetting score and moving avg. loss. Both methods outperform single model EL2N and surpass the random pruning baseline at low pruning rates on all datasets.

**EL2N Score Suffers from Randomness.** We observe that EL2N (1 model) under-performs on all three datasets (Fig. 4), being even worse than random pruning. The primary reason is that the EL2N score approximates the gradient norm of a sample at a certain epoch, and it assumes that the importance score of a sample remains static throughout the training process, ignoring the impact of evolving training processes. Disregarding the training dynamics leads to unreliable estimations that are highly susceptible to random variations. By training multiple models (20 models) and using the mean E2LN score, the performance improves significantly. However, it comes at the price of a notable growth in training time: it necessitates 20 times the training time required by our method.

**Differences in Datasets.** We notice considerable variations in the levels of sample redundancy across the three ReID datasets (Fig. 4). Using our method only allows for the removal of 5% of samples from Market1501 without compromising accuracy. While on MSMT and VeRi datasets our method can eliminate 15% and 30% of the samples, respectively. We hypothesize this is due to the rigorous annotation process of Market1501 dataset. Additionally, Market1501 is considerably smaller in scale, i.e., Market1501 training set is merely 43% the size of MSMT17 and 34% the size of VeRi, thereby reducing the likelihood of redundant or easy samples.

**Generalization from ResNet50 to Deeper ResNet and ViT.** Previous experiments have demonstrated that our proposed metric can better quantify the importance of samples. Next, we validate if the ordering (ranking) obtained using a simpler architecture, i.e., ResNet50 (He et al., 2016), can also work with a more complex architecture, i.e., deeper ResNet and ViT (Dosovitskiy et al., 2021). To this end, we train a ResNet101 and a vision transformer ViT-B/16 on a pruned dataset using the sample ordering from ResNet50. We plot generalization performance of ResNet101 and ViT in Fig. 5. We observed that the sample ordering obtained using ResNet50 is still applicable to ResNet101 and ViT. For instance, using ResNet101, 15% of MSMT17 samples can be removed, whereas with ViT, 30% of the VeRi samples can be eliminated without any reduction in accuracy. These results verify that our metric can reflect the ground-truth characteristics of samples, independent of the model architecture used for training.

### 5.3.2 Data Pruning on Classification Dataset

To comprehensively verify the effectiveness of our proposed metric, we conduct supplementary experiments on two classification datasets. We demonstrate the efficacy of our method on CIFAR-100 (Krizhevsky et al., 2009) and CUB-200-2011 (Wah et al., 2011) datasets and compare our method with the forgetting (Toneva et al., 2018) and EL2N (Paul et al., 2021) scores in Fig. 6. For a fair comparison, all methods employ the same training time or cost to estimate the importance scores of samples. In detail, following the standard protocol

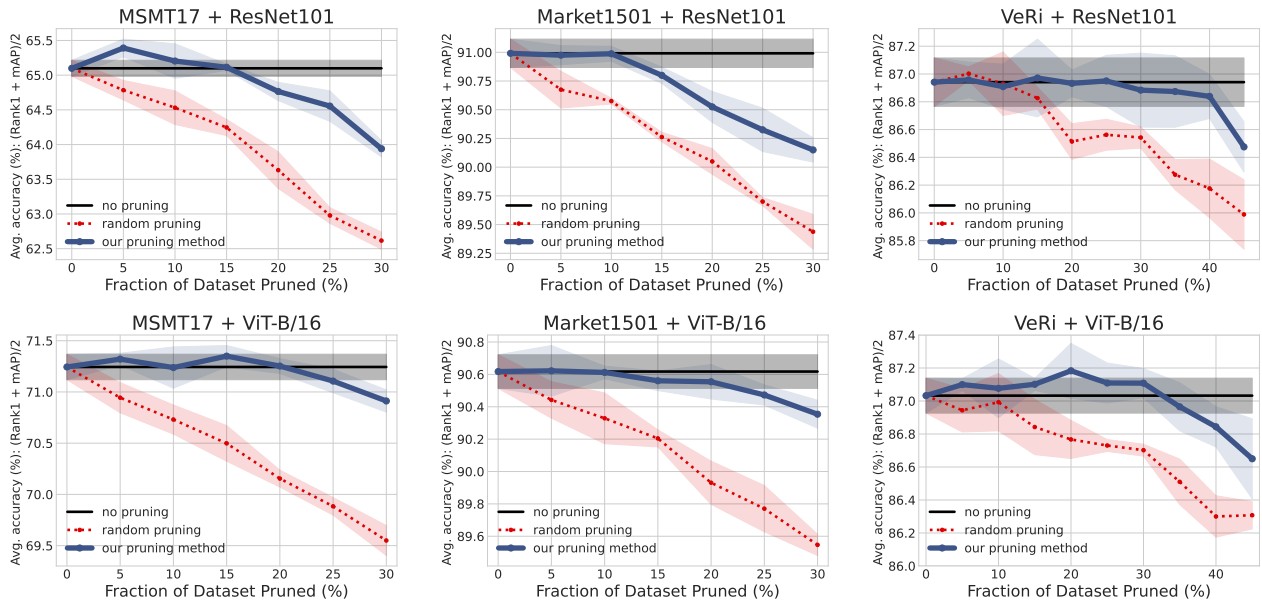

Figure 5: Generalization performance. We train a ResNet101 and a ViT model using the **sample ordering of ResNet50** on the MSMT17 dataset. For each method, we carry out four independent runs with different random seeds and we report the mean values. Shaded areas mean +/- one standard deviation of four runs.

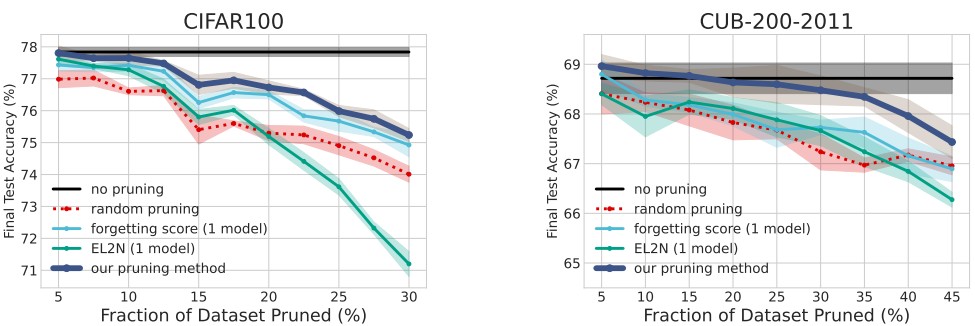

Figure 6: Data pruning on classification datasets. We report the mean of final test accuracy on two classification datasets (labeled at the top), obtained by training on the pruned datasets. For each method, we carry out four independent runs with different random seeds and report the mean.

of EL2N (Paul et al., 2021) we solely train a model for 20 epochs on CIFAR-100 and 3 epochs on CUB-200-2011, equivalent to 10% of all training epochs, and then apply three methods, i.e., forgetting, EL2N and our scores to estimate the sample importance. In line with the results in ReID tasks, our method consistently demonstrates superior performance over two competing methods. Notably, we intentionally exclude pre-training methodologies for both classification experiments; specifically, the model weights are randomly initialized. These results confirm the consistent efficacy of our approach, regardless of the utilization of pre-trained models.

## 5.4 Robust Training on Noisy Datasets

We test the efficacy of our data purification method on random noisy datasets. Herein, we start with the original images with the assigned labels. A certain percentage (i.e., from 10% to 50%) of training images are randomly selected and intentionally assigned the wrong labels. We compare our approach with a state-of-

the-art method for identifying mislabeled samples, i.e., AUM (Pleiss et al., 2020), and evaluate each of our used components, i.e., label correction and removing outliers.

From Fig. 7, we observe that label correction plays a pivotal role, while merely removing outliers only slightly improves the performance. The reason is that outlier removal merely discards a few data points, leaving behind a substantial amount of mislabeled data, which ultimately results in minimal model improvement. Besides, we observed these two components are interdependent and complementary to each other, and the simultaneous utilization of them significantly boosts the performance.

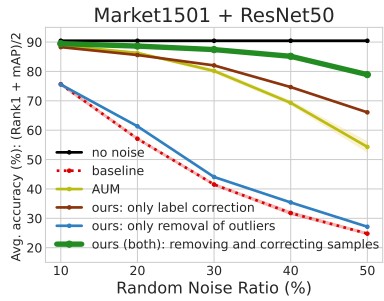

Our proposed data purification approach, *removing and correcting samples*, demonstrates superior performance compared to the AUM metric (Pleiss et al., 2020). The rationale behind this advantage lies in the fact that the AUM metric tends to discard a greater amount of training data, whereas our method selectively removes fewer samples and capitalizes on mislabeled data by employing label correction to maximize the benefit of each sample.

Figure 7: Data purification on noisy datasets. We report accuracy under different random noise ratios. Each curve represents the mean accuracy over four independent runs.

## 5.5 Putting It All Together

By integrating data purification we build a comprehensive data pruning approach. Samples are removed or rectified according to the sequence below: 1) all outliers (i.e., samples whose highest class score is lower than 10%) are removed; 2) all samples with incorrect labels are rectified; 3) easy samples (i.e., samples whose soft labels have low entropy) are pruned. Please note that the total number of pruned samples equals the sum of *all* outliers and the pruned easy samples (i.e., all outliers are removed). We test our approach on three ReID datasets and evaluate the benefit of our data purification approach. In Fig. 8, we observe marked performance differences between datasets. Data purification on MSMT17 shows a notable effect, potentially due to the presence of more outliers and mislabeled samples in this dataset. The performance on VeRi is slightly improved as well. However, the impact on Market1501 is comparatively limited, plausibly owing to stricter annotation protocols and fewer outliers compared to the other datasets. In summary, the seamless integration of data purification and data pruning can effectively boost overall performance. Especially, on MSMT17 our approach can even eliminate/reduce 30% of samples/training time with almost no compromise in performance.

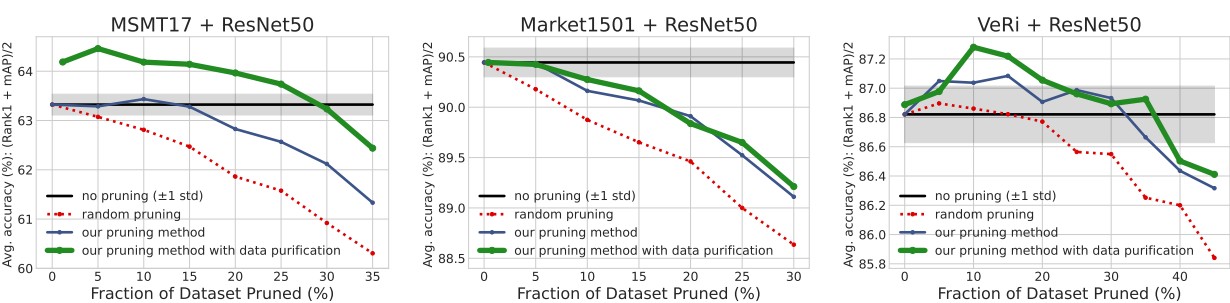

Figure 8: Data pruning with data purification. Accuracy is achieved by training on the pruned dataset (pruning ratios on the X-axis). We report the mean values from four independent runs with different seeds.

## 5.6 Sensitivity Analysis and Ablation Studies

Our approach involves two hyper-parameters: (i) the number of training epochs required to generate the soft labels, i.e., $T$ from Eq. 1, and (ii) the threshold $\delta$ to remove outliers. We *first* explore the impact of the

hyper-parameter $T$. *Next*, we verify the importance of logit accumulation through ablation experiments, and *finally* analyze the effect of the threshold $\delta$ on the performance of data purification.

**Impact of Score Computation Epoch $T$.** We investigate how early in training our metric is effective at estimating the importance scores of samples. In Fig. 9, we compare the model performance from training on 85% training data but pruned based on the forgetting score (Toneva et al., 2018), EL2N score (Paul et al., 2021) and our scores computed at different epochs. All scores are estimated using the identical random seed. We observe that after 12 epochs, the model performance of using our pruning method essentially stabilizes and surpasses that of random pruning as well as other competing methods. Considering EL2N's vulnerability to randomness, a single model is often insufficient to precisely estimate sample importance. The forgetting score typically necessitates many training epochs (Toneva et al., 2018), thereby hindering its capability to estimate sample importance in early training stages accurately. Additionally, we explore the impact of score computation epoch $T$ on label correction and present results in Appendix D.1. We observe training for 12 epochs is sufficient to achieve desirable results of label correction as well.

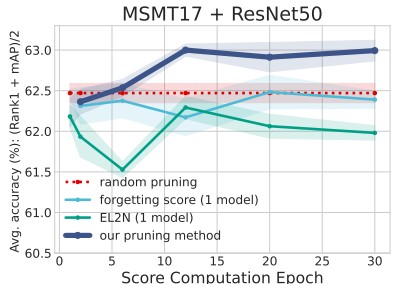

Figure 9: Impact of score computation epoch $T$. Model performance achieved by training on 85% training data comprised of examples with maximum forgetting, EL2N, and our scores computed at different epochs.

**Importance of Logit Accumulation.** We conduct two experiments to demonstrate the importance of logit accumulation in (i) importance score estimation for data pruning, and (ii) label correction on noisy datasets. In both experiments, our soft labels are generated using different frequencies of logit accumulation, i.e., once at the $12^{th}$ epoch and every 1-6 epochs. For example, *"every 4 epochs"* means we use logits at the $4^{th}$, $8^{th}$, and $12^{th}$ epochs to calculate our proposed importance scores. Likewise, *"every 6 epochs"* means logits are accumulated at the $6^{th}$ and $12^{th}$ epoch. Figure 10 depicts the results of data pruning experiment, and we observe the progressive improvement in pruning performance with increasing frequency of logit accumulation. Consistent with this result, Figure 11 illustrates that as more logits at different epochs are accumulated, the performance of label correction improves. Both results validate our claim that the soft label generated using the full logit trajectory can better capture the ground-truth characteristics of a sample.

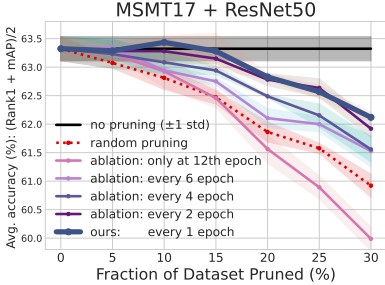

Figure 10: Ablation study of logit accumulation for *data pruning*. Model performance when trained with the importance scores computed using different frequencies of logit accumulation.

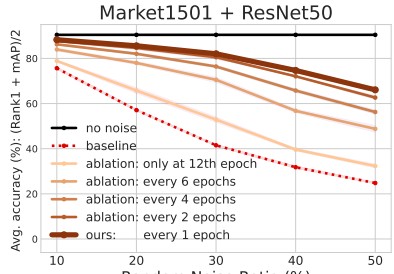

Figure 11: Ablation study of logit accumulation on *noisy datasets*. Model performance *only* using label correction and the soft labels are generated using different frequencies of logit accumulation.

**Impact of Outlier Removal Threshold $\delta$.** In our method, samples whose highest class score is lower than $\delta$ are marked as outliers and removed (Sec. 4). We perform the ablation study on Market1501 dataset to quantify the impact of $\delta$ under varying levels of noise. We keep the noise ratios fixed at 10%, 20% and

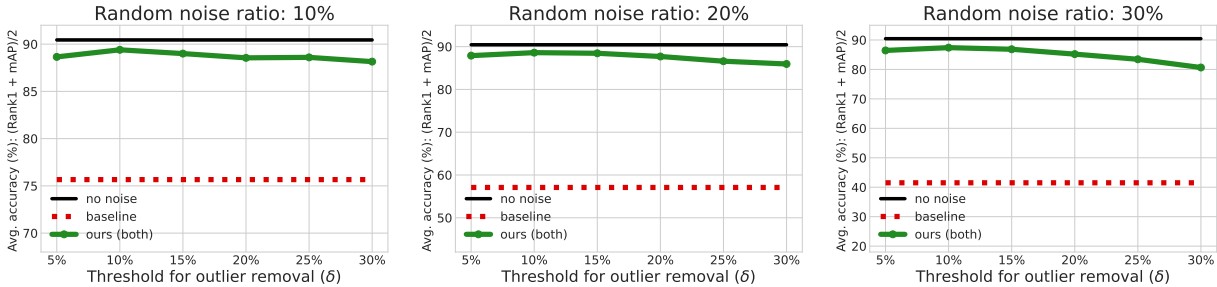

Figure 12: Impact of outlier removal threshold $\delta$. Model performance under noise ratio 10%, 20% and 30%.

30% while varying $\delta$. As observed in Fig. 12, setting $\delta = 10\%$ yields slightly superior performance. Overall, $\delta$ has a limited impact on the final performance, especially when $\delta$ lies between 5% and 15%. It demonstrates that our approach is robust and insensitive to the threshold values.

## 6    Discussion and Conclusion

In this work, we have addressed two issues of ReID datasets (less informative samples and noise) by proposing a plug-and-play architecture-agnostic data pruning framework. *Firstly*, by leveraging the training dynamics, we have provided a more accurate and robust pruning metric with extremely low computational overhead. In the generalization test, we have demonstrated that our metric reflects the ground-truth characteristics of samples independently of the model architecture used for training, i.e., the sample ordering (ranking) obtained via a ResNet remains effective in training a vision transformer. For completeness, we have also verified our method on two image classification datasets, observing that our approach exhibits superior performance over competing methods as well. *Secondly*, we have proposed an efficient data purification method, enabling the correction of mislabeled samples and the removal of outliers. Experiments on noisy datasets validate that our data purification method exhibits robustness, achieving remarkable results. *Finally*, by integrating data purification we have built a comprehensive data pruning framework and demonstrated that it achieves state-of-the-art performance when pruning ReID datasets, allowing for the removal of 35%, 30%, and 5% of samples from the VeRi, MSMT17, and Market1501 datasets respectively, thereby leading to an equivalent reduction in training time. Like previous works (Toneva et al., 2018; Paul et al., 2021), our data pruning method requires labels. In the future, we intend to investigate data pruning methods based on logit trajectories in an unsupervised setting.

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

# A    Implementation Details for ReID Tasks

## A.1    Estimating the Importance Score of Samples from ReID datasets

All experiments are implemented in PyTorch (Paszke et al., 2019) using the FastReID (He et al., 2020) toolbox[1] on 4 NVIDIA Tesla V100 GPUs. A general workflow for supervised ReID is shown in Fig. 13. In our work, we follow the training procedure[2] of Luo et al. (2019). For feature extraction, we employ a ResNet50 (He et al., 2016) pre-trained on ImageNet (Deng et al., 2009); the model is trained for 12 epochs. All images from Market1501 and MSMT17 are resized to $256 \times 128$ pixels, while images from VeRi are resized to $256 \times 256$ pixels. During training, we record the logits for each sample after each forward pass. For optimization, we use a combination of the cross-entropy loss and the triplet loss, i.e., $L_{\text{train}} = L_{\text{ce}} + L_{\text{triplet}}$.

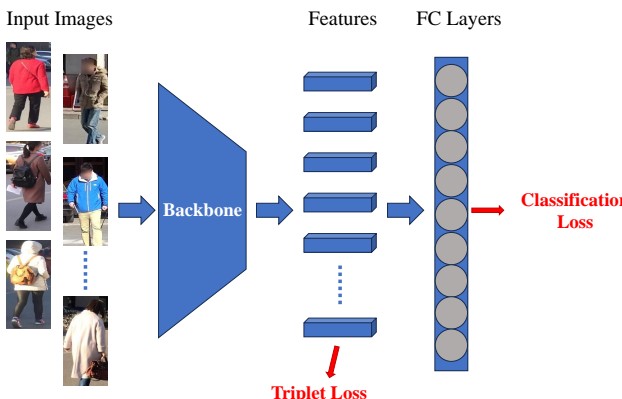

Figure 13: A general supervised ReID workflow.

## A.2    Existing Methods

**E2LN.**    We use the same model configuration and settings as in our approach, except for the loss function. When calculating the EL2N score, we strictly follow its procedure and definition (Paul et al., 2021), i.e., we only use the cross-entropy loss and do not utilize additional loss functions (e.g., metric loss). We train till 12 epochs and then compute the EL2N score for each sample. For EL2N (20 models), we run the same experiments 20 times but with different random seeds and obtain the score by averaging over the runs.

**Forgetting Score.**    We train a single model for 120 epochs using the default settings[2]. During training, we record the number of forgetting events for each sample.

**Supervised Prototypes.**    To ensure a fair comparison with other methods, we replace the self-supervised learning proposed in Sorscher et al. (2022) with supervised learning. Using the default settings and configuration[2], we train a single model for 120 epochs. After training, we calculate the supervised prototype score of each sample, which is defined by the L2 distance between a sample and its class centroid (Sorscher et al., 2022). Additionally, following Sorscher et al. (2022), we also apply a simple 50% class balancing ratio.

## A.3    Data Pruning Experiments on ReID datasets

Following the default model configuration and settings[2], we train four independent ReID models using different random seeds on the pruned dataset, where the samples are pruned based on different types of importance score, i.e., the forgetting, EL2N, supervised prototype and our scores. We then report the mean performance across these four models.

---

[1]https://github.com/JDAI-CV/fast-reid
[2]https://github.com/JDAI-CV/fast-reid/blob/master/configs/Base-bagtricks.yml  -  Model configuration and hyper-parameter setting.

### A.4 Generalization from ResNet to ViT

We initialize a vision transformer ViT-B/16 (Dosovitskiy et al., 2021) with its weight pre-trained on ImageNet21K and use the default configuration[3]. The model is trained on a pruned dataset, which is pruned based on the ranking list of samples generated by a ResNet50. We report the average performance across four runs with different random seeds.

### A.5 Robust Training on Noisy Datasets

To evaluate the performance of AUM, we follow their proposal (Pleiss et al., 2020) to fine-tune the threshold value using threshold samples. For our method, we set the threshold of outlier removal to $\delta = 10\%$ on all three ReID datasets. After the outlier removal and label correction, we train four ReID models using the default settings[2] and present the mean accuracy derived from these four runs.

### A.6 Putting It All Together

In our proposed framework, samples are either removed or rectified according to the sequence below:

1. Eliminating *all outliers* according to the highest class scores of the samples, as described in Sec. 4.

2. Removing easy samples, as described in Sec. 3.3.

3. Rectifying mislabeled samples. Details are presented in Sec. 4.

Please note that the total number of pruned samples equals the sum of all outliers and the pruned easy samples (i.e., all outliers are removed). For instance, if we set the threshold $\delta$ to 10% the noisy sample rate of the MSMT17 dataset is 1.1%. If 10% of the samples are removed, it includes 1.1% outlier samples and 8.9% easy samples.

## B Implementation Details for Classification Tasks

For image classification on CIFAR-100 and CUB-200-2011 datasets, we begin with computing the forgetting score (Toneva et al., 2018), EL2N score (Paul et al., 2021), and our proposed metric for each sample. For a fair comparison, all methods employ the same training time or cost to estimate the importance scores of samples. For experiments on CIFAR-100, we use the model architecture and training parameters[4] specified in Toneva et al. (2018), while the model architecture and training parameters[5] specified in Boudiaf et al. (2020) are used for experiments on CUB-200-2011. Following the standard protocol of Paul et al. (2021), we solely train a model for 20 epochs on CIFAR-100 and 3 epochs on CUB-200-2011, equivalent to 10% of all training epochs. After computing the individual metrics for each sample, we remove the corresponding number of easy samples from the original training set. For instance, if the pruning rate is 20%, we remove the top 20% of the easiest samples (samples with the lowest importance score). Following Toneva et al. (2018), we train four models *from scratch* (i.e., weights are randomly initialized without the use of any pre-trained model) on this pruned dataset independently. For each run, we use a different random seed. Finally, we report the mean test accuracy across these four independent runs.

## C Examples of different outliers from MSMT17.

In Fig. 14 we present three different types of outliers from MSMT17, i.e., heavy occlusion, multi-target coexistence and object truncation

---

[3]https://github.com/JDAI-CV/fast-reid/blob/master/configs/Market1501/bagtricks_vit.yml - ViT configuration

[4]https://github.com/mtoneva/example_forgetting

[5]https://github.com/jeromerony/dml_cross_entropy

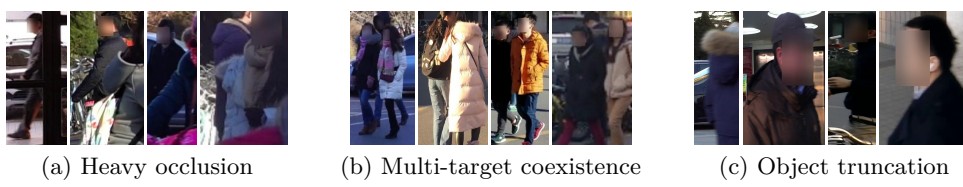

(a) Heavy occlusion   (b) Multi-target coexistence   (c) Object truncation

Figure 14: Examples of different outliers from MSMT17.

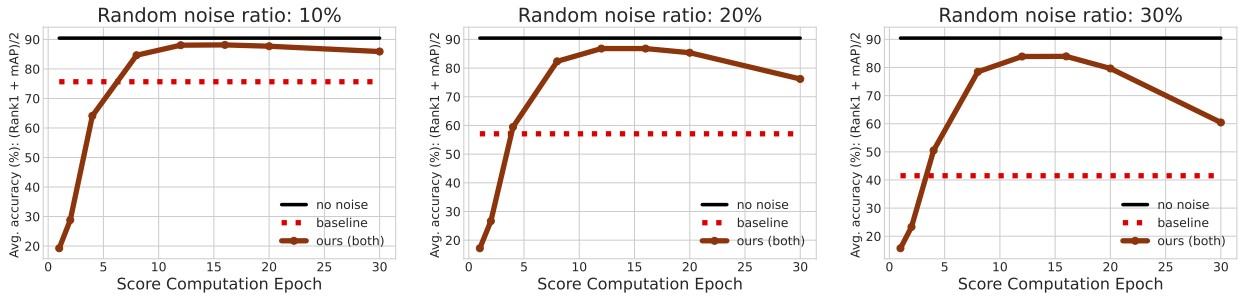

Figure 15: Accuracy achieved by training on the Market1501 dataset under different noise ratios *only using label correction*, where the soft labels are generated at different epochs.

# D Ablation Study

## D.1 Impact of Score Computation Epoch *T* on Label Correction

In Sec. 5.6, we investigate how early in training our metric is effective at estimating the importance scores of samples and observe that after 12 epochs, the importance scores of samples estimated by our method tend to be stabilized. However, it beckons the consideration: Are the generated soft labels trained for 12 epochs sufficient to achieve a good performance in label correction? To answer this question, we explore the impact of score computation epoch $T$, which is the soft labels generation epoch as well, on label correction. We present results in Fig. 15. Our sensitivity analysis reveals that the soft labels generated between the $8^{th} - 20^{th}$ epochs demonstrate a superior label correction capability, even approaching the model performance trained on the clean dataset. This observation clarifies our earlier question: Even in the relatively early stages of training (after 8 epochs), the soft labels we proposed still exhibit a notable capability of label correction.

Furthermore, we observe that the label correction capability of soft labels generated after 20 epochs exhibits a decline, particularly pronounced under high noise ratios such as 30%. The underlying rationale is that, as the model undergoes prolonged training on a noisy dataset, the model starts memorizing a huge amount of mislabeled samples, leading to fitting these erroneous labels (Pleiss et al., 2020). Overall, our approach exhibits robustness to hyper-parameter $T$, particularly within an acceptable range of noise ratios., e.g. 10%, 20%.

# E Example Images

We present some samples from three ReID datasets and CIFAR-100 dataset sorted from easy to hard based on our proposed pruning metric. From Figs. 16 to 19, we observe that the samples with the smallest importance scores tend to be simple and are canonical representations of each class. In contrast, the samples with higher scores are harder to identify: they have different backgrounds or even suffer from occlusion or truncation.

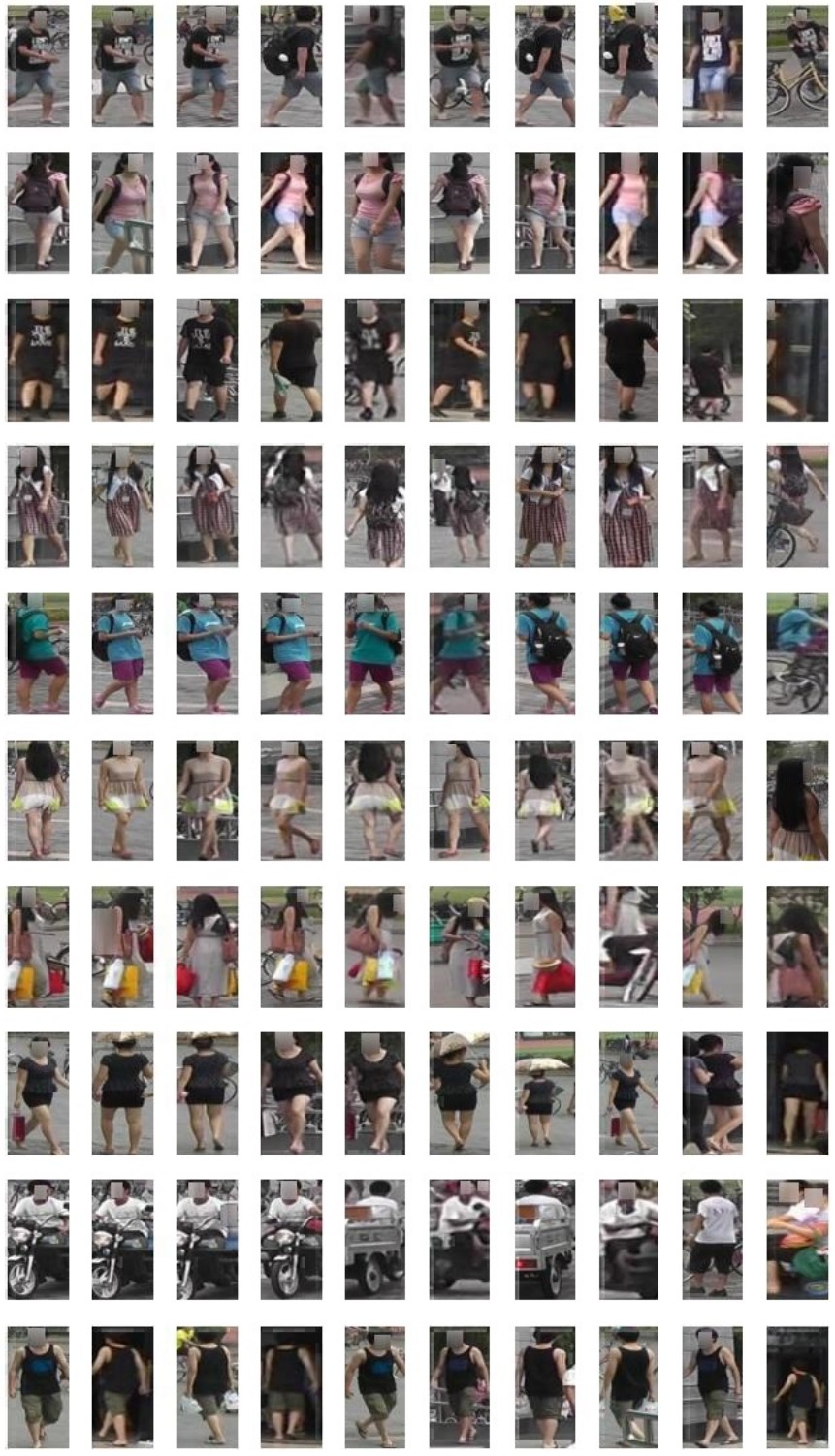

Figure 16: Samples from Market1501 sorted from *easy* (the first column) to *hard* (the last column) based on our proposed importance scores. Images in each row belong to the same identity.

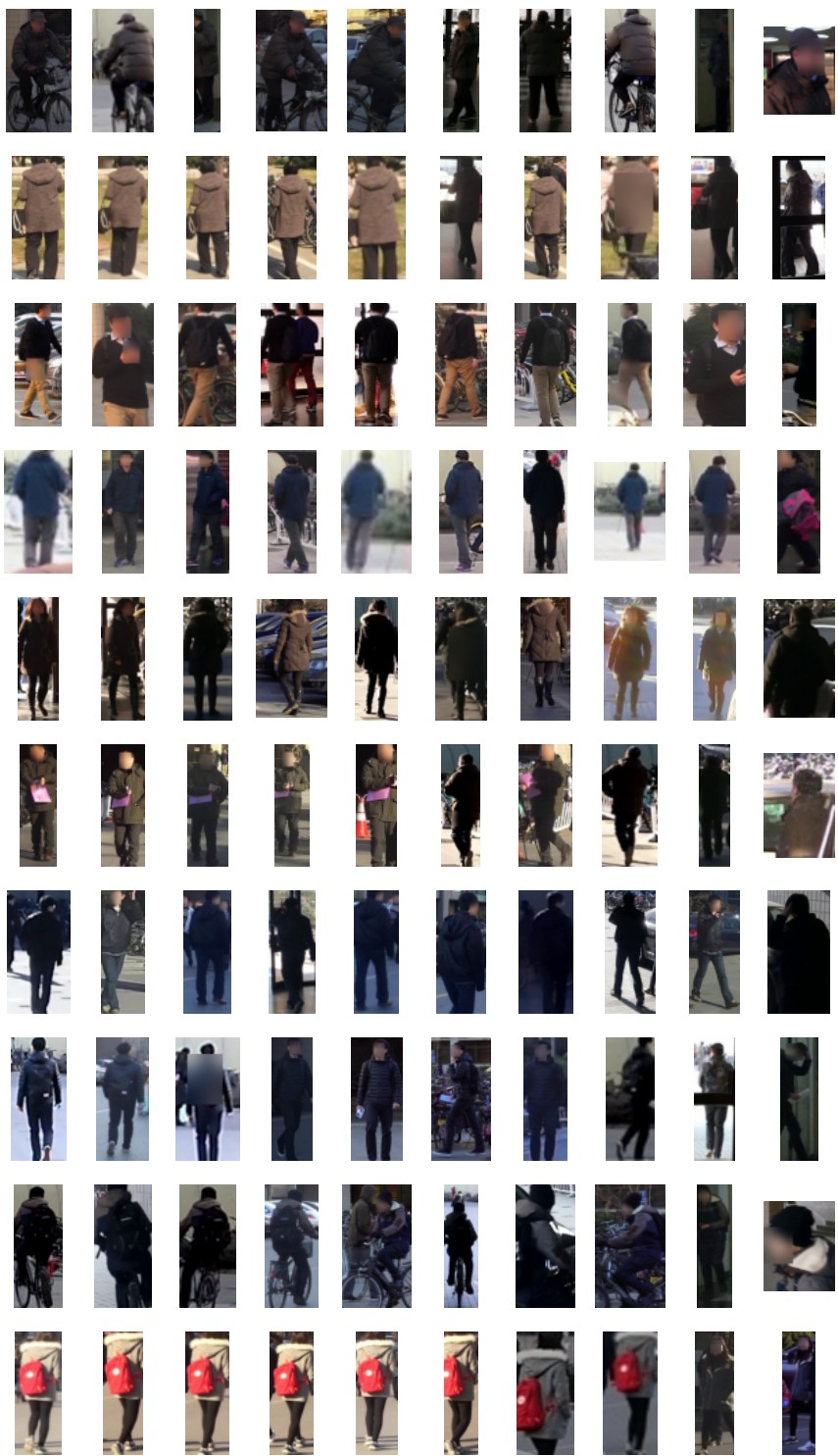

Figure 17: Samples from MSMT17 sorted from *easy* (the first column) to *hard* (the last column) based on our proposed importance scores. Images in each row belong to the same identity.

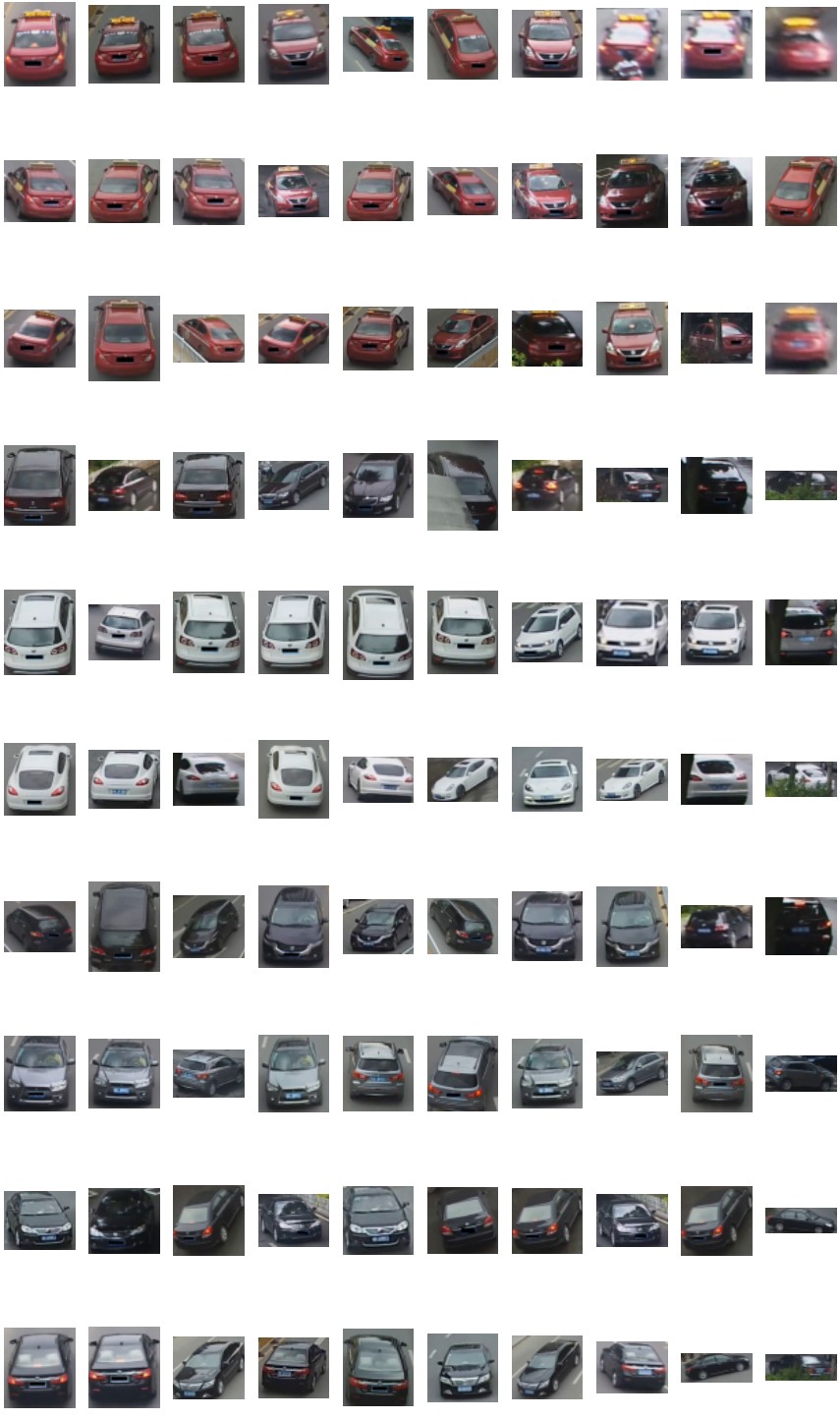

Figure 18: Samples from VeRi sorted from *easy* (the first column) to *hard* (the last column) based on our proposed importance scores. Images in each row belong to the same identity.

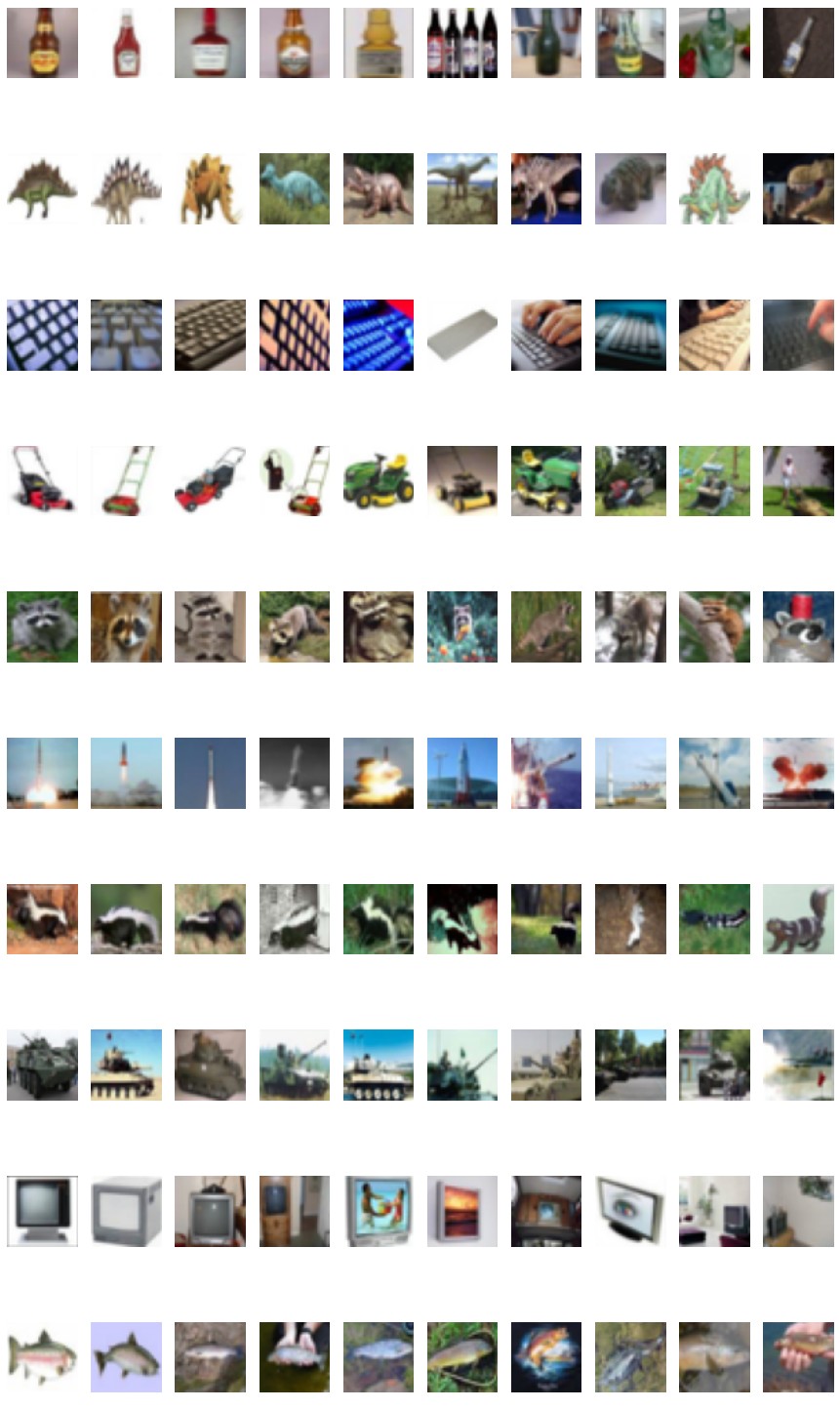

Figure 19: Samples from CIFAR-100 sorted from *easy* (the first column) to *hard* (the last column) based on our proposed importance scores. Images in each row belong to the same identity.

