# OpenReview forum: "Data Pruning Can Do More: A Comprehensive Data Pruning Approach for Object Re-identification"
_TMLR — Accepted by TMLR_

### Review · Reviewer_H1m6 · 2024-01-30

**Summary Of Contributions:**

The paper presents a data pruning strategy for training an object re-identification model. The proposed method leverages the logit trajectories of each training data instances and constructs a soft label based on the trajectory average. The soft label and its entropy are then used to perform data pruning, correcting incorrect labels or identifying outliers. The method is evaluated on three RID benchmarks using ResNet50/ViT-B/16 backbones.

**Audience:**

Yes

**Broader Impact Concerns:**

No concerns.

**Claims And Evidence:**

Yes

**Requested Changes:**

1. Clearer motivation for the RID task.
2. Comparisons with previous work on data selection in training.
3. As mentioned above, more comprehensive experimental evaluation.

**Strengths And Weaknesses:**

Strengths:

1. The problem of data pruning in the RID task seems reasonable and its solution is useful in practice.

2. The experiments show some promising results on three benchmarks under its model setting.

Concerns:

1. The connection between the proposed data pruning strategy and the RID task is weak. It lacks specific considerations for the RID task -- the entire design is based on the standard classification paradigm. As such, it is unclear why such a generic method is only evaluated on the RID task.

2. Data selection is not a new problem for learning from data with noisy labels. The idea of using training dynamics have been explored before, which has not been discussed in the paper, for example:
   Tianyi Zhou, Shengjie Wang, and Jeff Bilmes. Robust curriculum learning: from clean label detection to noisy label self-correction. In International Conference on Learning Representations, 2021.

3. The technical contribution is incremental: it is rather heuristic to use the averaged logits or its entropy for various data cleaning steps, and no theoretical analysis has been provided. What is the difference between the RID training and the standard classifier training in this problem setting?

4. The experimental evaluation is less convincing in several aspects:

    a) As an empirically design strategy, more extensive experiments on different settings of backbone networks should be reported, such as deeper ResNet and larger ViT models, with different pre-training settings.

    b) The efficiency achieved by such data pruning is underwhelming. At most 30% of the dataset can be removed. It should be compared with other few-shot or fine-tuning learning strategies regarding training efficiency.

    c) The baseline methods are weak. As mentioned above, it is more convincing to compare this method to data selection strategies used in learning with noisy labels, which also ignore "hard examples" based on small loss heuristics.

---

> ### Author Response · Authors · 2024-03-11
>
> **First and foremost**, we are grateful for the valuable suggestions and feedback from the reviewer. Below is our response to the specific concerns raised:
>
> ##  Responses to Weaknesses:
> ### (1) The connection between our data pruning strategy and the ReID task.
> **Response:** We appreciate the reviewer for pointing out the concern regarding the connection between our method and ReID. Our approach is indeed motivated by ReID tasks, specifically the observation of a significant number of similar or mislabeled samples in ReID datasets. However, these issues are not limited to ReID tasks and also occur in classification tasks. Consequently, we explore whether data pruning methods designed for classification tasks can be applied to ReID tasks. Then, we introduce a widely applicable metric suitable for both ReID and classification tasks and validate its performance on three ReID and two classification datasets (CIFAR100 and CUB-200). Our work is dedicated to bridging the gap in applying (general) data pruning to ReID datasets.
>
> ### (2) The comparison between our approach and [1].
> **Response:** We are grateful to the reviewer for providing relevant literature. However, the work of [1] focuses on dynamic sample selection rather than data pruning. Dynamic sample selection chooses samples in real-time during training to enhance learning efficiency without reducing the overall dataset size. In contrast, data pruning is a static process of permanent removing less useful samples before training to improve the training efficiency and it is worth to note that a dataset only needs to be pruned once and then can be used for training with different architectures in the future without the need for further pruning.
> To fairly compare with [1], we utilized its core idea of moving average loss to realize static data pruning. The results are presented in Fig. 4, indicating that our approach consistently outperforms theirs across three ReID datasets. The primary reason is that the work of [1] uses CE loss as loss function, which only captures the target-class-related knowledge (only measuring the discrepancy between the predicted probability and the true probability of the target classes) while ignoring the information across non-target classes. However, knowledge from non-target classes can potentially be of greater importance than that from the target class, as demonstrated in [2]. Different from [1], our method utilizes information from all classes (logits of all classes) to present a more comprehensive metric, resulting in superior results.
>
> ### (3) Theoretical analysis and the difference between ReID training and classifier training in our problem setting.
> **Response:** Our paper is mainly an empirical study. However, we can get some valuable insights from these experiments – utilization of training dynamics (i.e., logit accumulation) provides a better and more robust metric to evaluate the importance of a sample. We consider the ReID task as a specialized form of classification task, so when applying our data pruning method, there is no difference between ReID and standard classification tasks during the training stage.
>
> ### (4a) Applied to deeper or different network architectures.
> **Response:** We are grateful to the reviewer for their valuable suggestion regarding the experiments. We validated our method on a deeper network architecture (ResNet101). The results, as depicted in Fig. 5, align with our expectations. We observed that the sample importance ranking derived from a smaller model still performs remarkably well on the larger model. Furthermore, it is worth noting that the majority of ReID models apply ImageNet pre-trained model and use ResNet50, ResNet101 or ViT as backbone [3-6]. Thus, our experiments have covered most commonly employed models/backbones to validate the effectiveness of our method.
>
> ### (4b) The efficiency of our method and compare with few-shot or fine-tuning strategies.
> **Response:** Our results have significantly outperformed the SOTA data pruning methods. Under the same computational costs, our method can permanently remove 15% of samples from MSMT17, while other SOTA data pruning methods (e.g., EL2N) can only remove 5% of samples. Our method can remove two times more samples.
>
> Our methods are orthogonal to few-shot or fine-tuning strategies. After data pruning using our method, few-shot or fine-tuning can still be applied in order to further improve the training efficiency.
>
> ### (4c) The baseline methods are weak.
> **Response:** The primary focus of our paper is the application of existing and proposed data pruning methods to the ReID task. Consequently, we compared three SOTA data pruning methods on ReID datasets. Additionally, for comprehensive comparison, we compared the method, that the reviewer mentioned. Results are illustrated in Fig. 4. Our approach still provides superior results.

---

> > ### Author Response · Authors · 2024-03-11
> >
> > **References:**
> >
> > [1] Tianyi Zhou, Shengjie Wang, and Jeff Bilmes. Robust curriculum learning: from clean label detection to noisy label self-correction. In International Conference on Learning Representations, 2021.
> >
> > [2] Borui Zhao, Quan Cui, Renjie Song, Yiyu Qiu, and Jiajun Liang. Decoupled knowledge distillation. In Proceedings of the IEEE/CVF Conference on computer vision and pattern recognition, pp. 11953–11962, 2022.
> >
> > [3] Zhedong Zheng, Xiaodong Yang, Zhiding Yu, Liang Zheng, Yi Yang, and Jan Kautz. Joint discriminative and generative learning for person re-identification. In CVPR, 2019. 2, 5, 6
> >
> > [4] Lingxiao He, Xingyu Liao, Wu Liu, Xinchen Liu, Peng Cheng, and Tao Mei. Fastreid: A pytorch toolbox for general instance re-identification. Proceedings of the 31st ACM International Conference on Multimedia. 2023.
> >
> > [5] Guan’an Wang, Shuo Yang, Huanyu Liu, Zhicheng Wang, Yang Yang, Shuliang Wang, Gang Yu, Erjin Zhou, and Jian Sun. High-order information matters: Learning relation and topology for occluded person re-identification. In CVPR, pages 6449–6458, 2020. 8
> >
> > [6] Guanshuo Wang, Yufeng Yuan, Xiong Chen, Jiwei Li, and Xi Zhou. Learning discriminative features with multiple granularities for person re-identification. In ACMMM, pages 274–282, 2018. 1, 2, 3, 8

---

### Review · Reviewer_xbcP · 2024-02-06

**Summary Of Contributions:**

The paper addresses efficient training by discarding or pruning less important data from re-identification (re-id) datasets. The proposed approach, unlike previous approaches, not only prunes less informative data points, but also can discard outlier as well as rectify noisy labels. The approach is built on top of the intuition that a hard (informative) example remains a tough nut for most of its journey throughout the training. Mathematically, the authors measure this by taking average of the logits of such examples across different epochs and computing the entropy of this. This gives the importance score that forms the basis to categorize the example as hard or easy and also as noisy or not. The authors took re-id as the use case and showed the efficacy of the approach on re-id datasets along with well designed and executed ablation studies.

**Audience:**

Yes

**Broader Impact Concerns:**

None.

**Claims And Evidence:**

Yes

**Requested Changes:**

The requested changes are mentioned in the weaknesses. I will be happy to change my rating on seeing these.

**Strengths And Weaknesses:**

Strengths:
1. The proposed computation of soft labels as average logits over training epochs and the importance score as entropy of these soft labels is simple, intuitive and effective. It is also straightforward a metric as mentioned by the authors.
2. Comprehensive experiment is one of the major strengths of the work. The approach shows its superiority on 3 re-id and one typical image classification dataset over a few state-of-the-art data pruning approaches. Leveraging the training dynamics helps not only to prune effectively for training in convnets but the pruned dataset is shown to be effective in ViT also. Another interesting experiment is label correction and outlier removal on noisy datasets. Sensitivity analysis on the hyperparameters $T, \delta$ and frequency of logit accumulation are also worth mentioning.

Weaknesses:
1. The work looks related to coreset selection [a, b] and dataset distillation [c, d] in flavor (especially if I look at the goal of training with only a handful of important data). A discussion relating the proposed work to these types of works substantiating the similarities and differences would be good to have.
2. Though the target scenario for the work is re-id, it would be good to apply the proposed approach to a largescale typical image classification dataset e.g., ImageNet or the fine grained classification dataset CUB-200. Though, experimental results show little superiority of the proposed approach on a small scale CIFAR-100 dataset, it would be good to have such experiments along with it a discussion on the approach’s success or failure as the case may be.
3. Some minor typos:
 - Related work: First line – the word ‘as’ will not be there.
 - Section 4: “Dataset Noise” Last line: ‘eliminate’ -> ‘eliminating’
 - Section 5.3.1: “The disregard of training dynamics” -> “Disregarding the training dynamics”

[a] K. Killamsetty, D. Sivasubramanian, G. Ramakrishnan, and R. Iyer, “Glister: Generalization based data subset selection for efficient and robust learning,”, AAAI, 2023.

[b] B. Mirzasoleiman, J. Bilmes, and J. Leskovec, “Coresets for dataefficient training of machine learning models,” ICML, 2020.

[c] T. Wang, J.-Y. Zhu, A. Torralba, and A. A. Efros, “Dataset distillation,” arXiv preprint arXiv:1811.10959, 2018.

[d] J. Cui, R. Wang, S. Si, C.-J. Hsieh, “Scaling Up Dataset Distillation to ImageNet-1K with Constant Memory”, ICML 2023.

---

> ### Author Response · Authors · 2024-03-11
>
> **First and foremost**, we sincerely appreciate the reviewer’s consideration of our work, including the acknowledgment of strengths and identification of weaknesses. We would like to address the concerns raised.
>
> ## Strength Acknowledgment:
> We are delighted that the reviewer recognizes our method for its simplicity and effectiveness. Furthermore, we are pleased with the reviewer’s acknowledgment of our experiments, especially the generalization test on ViT, sensitivity analysis, and ablation studies.
>
> ## Responses to Weaknesses:
> ### (1) The connection between coreset selection and dataset distillation
> **Response:** The work of [a] focuses on dynamic sample selection, which iteratively chooses different samples during the training process to enhance learning efficiency without reducing the overall dataset size. In contrast, data pruning permanently removes less useful data to reduce the dataset size and improve learning efficiency and it is worth noting that a dataset only needs to be pruned once and then can be used for training with different architectures in the future without the need for further pruning. Additionally, [a] requires solving an NP-hard problem every *L* epochs to find suitable samples. Although various approximations are used to address this NP problem, substantial computational effort is needed. Moreover, to identify the noisy sample it relies on a validation dataset, presuming it's entirely clean, whereas our method doesn't need an extra clean validation dataset.
>
> CRAIG [b] is similar to EL2N, both using gradients to assess sample importance. The key difference is that EL2N calculates the norm of the gradient for each sample at a specific epoch, whereas [b] computes the difference in gradients between samples. However, both methods overlook the impact of training dynamics on gradients, which evolve throughout the training process. To address this issue, our method employs the average logit to summarize the overall importance of samples throughout the training process.
>
> The studies in [c] and [d] concentrate on dataset distillation, aiming to synthesize a small and informative dataset to replace the larger original one. Nevertheless, due to computational constraints, these techniques typically synthesize a very small number of examples (e.g., 50 images per class) and the performance is far from satisfactory. Consequently, the performance of dataset distillation cannot be directly compared with that of dataset pruning.
>
> ### (2) Evaluation on more datasets and a discussion on the successes and failures
> **Response:** We appreciate the reviewer's suggestion to evaluate our method on more datasets and their interest in a deeper discussion on when our method succeeds and when it fails. As suggested by the reviewer, we conducted an experiment on the fine-grained classification dataset CUB-200. Figure 6 in the updated version shows the results: Same as the results of our previous data pruning experiments, we observed that our method still outperforms SOTA data pruning methods on CUB-200 dataset.
>
> Discussion on cases of success and failure: The effectiveness of our method largely depends on the quality or redundancy of a dataset. For instance, on the Market1501 dataset, we can remove only 5% of samples due to its rigorous annotation and sample selection process, whereas on the VeRi dataset, we can remove 35% of samples. In general, the more similar samples there are in a dataset, the better our method performs; conversely, the more the samples differ, the less effective our pruning method is.
>
> ### (3) Some minor typos
> **Response:** We are grateful to the reviewer for pointing out the typographical errors in our paper. We have undertaken a comprehensive review to ensure that all of these issues are corrected.
>
> **References:**
>
> [a] K. Killamsetty, D. Sivasubramanian, G. Ramakrishnan, and R. Iyer, “Glister: Generalization based data subset selection for efficient and robust learning,”, AAAI, 2023.
>
> [b] B. Mirzasoleiman, J. Bilmes, and J. Leskovec, “Coresets for dataefficient training of machine learning models,” ICML, 2020.
>
> [c] T. Wang, J.-Y. Zhu, A. Torralba, and A. A. Efros, “Dataset distillation,” arXiv preprint arXiv:1811.10959, 2018.
>
> [d] J. Cui, R. Wang, S. Si, C.-J. Hsieh, “Scaling Up Dataset Distillation to ImageNet-1K with Constant Memory”, ICML 2023.

---

### Review · Reviewer_26U1 · 2024-03-03

**Summary Of Contributions:**

This paper mainly targets data pruning. The authors analyze the current data pruning methods and point out that these methods cannot handle noisy samples and ignore the full training information. To solve these problems, they propose a new method which records the pseudo labels trajectory during training and prunes data according to the confidence calculated by entropy of pseudo labels. The authors further propose several strategy to use their method to rectify mis-labeled samples and identify outliers. They conduct extensive experiments on both object re-id and image classification to prove the effectiveness of the proposed method.

**Audience:**

Yes

**Claims And Evidence:**

Yes

**Requested Changes:**

Please refer to the weaknesses and provide corresponding analysis and results.

**Strengths And Weaknesses:**

Strengths:
1. The idea of using global training information is solid.
2. The proposed method is clearly described.

Weaknesses:
1. Although this paper focuses on data pruning for object re-id, it is not clear why the proposed method is specifically designed for object re-id.
2. In Sec.5.5 the authors use both easy sample and outlier pruning together. I wonder what exact strategy is used for these two methods. Does 10% prune rate mean deleting 5% easy samples and 5% outliers?
3. In the proposed method the authors take $\tilde{y}$ as ground truth pseudo labels which are used to justify whether a sample is mis-labeled. I wonder if there is any chance the wrong labels are easy to learn, which leads to $argmax(\tilde{y})=y$ while it is still mis-labeled?
4. It would be great if the authors could provide comparison among different model scales, such as ViT-S, ViT-B and ViT-L.
5. The authors can also provide visualization of detected easy/outlier samples in CIFAR-10.

---

> ### Author Response · Authors · 2024-03-11
>
> **Firstly**, we sincerely appreciate the reviewer's recognition of our approach in using full training dynamics to estimate sample importance. Below are our responses to each of the reviewer’s concerns:
>
> ## Responses to Weaknesses:
> ### (1) The connection between our method and ReID
> **Response:** We appreciate the reviewer for pointing out the concern about the connection between our method and ReID. It seems there may have been a lapse in our explanation, which led to the impression that our method is specifically tailored for ReID.
> Our approach is indeed motivated by ReID tasks, specifically the observation of a significant number of similar or mislabeled samples in ReID datasets. However, these issues are not limited to ReID tasks and also occur in classification tasks. Consequently, we explore whether data pruning methods designed for classification tasks can be applied to ReID tasks. Then, we introduce a widely applicable metric suitable for both ReID and classification tasks and validate its performance on three ReID and two classification datasets (CIFAR100 and CUB-200). Our work is dedicated to bridging the gap in applying (general) data pruning to ReID datasets.
>
> ### (2) How to prune easy samples and outliers together?
> **Response:** We are grateful to the reviewer for pointing out the unclear parts of our paper. In Sec.5.5, the total number of pruned samples equals the sum of all outliers and the pruned easy samples (i.e., all outliers are removed). For instance, the noisy sample rate of the MSMT17 dataset is 1.1%. If 10% of the samples are removed, it includes 1.1% outlier samples and 8.9% easy samples. It's worth noting that the noise rate of each of these three ReID datasets is below 1.1%.
>
> ### (3) Reliability of generated pseudo labels: potential discrepancies with ground truth labels?
> **Response:** At the very early stages of training, if two classes have extremely similar appearances, pseudo-labels may mistakenly group them as one. This is observed in Fig.15 (Appendix), where the accuracy is notably low after training only for 5 or fewer epochs (4% of total training epochs).
> However, thanks to the idea of avg. logit, at a low noise rate (10%), the accuracy stabilizes at a high level after training for more than 8 epochs. With a high noise rate (greater than or equal to 30%), the accuracy also reaches a significantly high level after 8 to 20 epochs of training. In summary, our experiments demonstrate that our approach is relatively effective in addressing this problem.
>
>
> ### (4) Comparison among different model scales
> **Response:** In line with the suggestion from the reviewer H1m6, we extended our generalizability test to ResNet101 and the results are presented in Fig. 5. Our current experiments involve three networks: ResNet50, ResNet101, and ViT-B. In our generalizability test, we utilize the small size model (ResNet50) to create the ranking of sample importance and to prune easy samples, thereafter validating this ranking on a larger model (ResNet101) and a different model architecture (ViT-B). Our observation confirms that our metric reliably reflects the intrinsic characteristics of samples, independent of the model architecture and size utilized for training. Specifically, the sample ranking achieved by ResNet50 remains valid for training using ResNet101 or Vision Transformer.
> Due to hardware limitations, we cannot provide results for ViT-Large immediately. Nevertheless, based on our experiments with ResNet, we believe our method can generalize well to ViT-Large and ViT-Huge.
>
> ### (5) Visualization of detected easy/outlier samples in CIFAR-100
> **Response:** We are thankful for this suggestion. The visualization of easy and outlier samples from CIFAR-100 is depicted in Fig 19.

---

### Review · Reviewer_8NyC · 2024-03-03

**Summary Of Contributions:**

This work makes the observation that not all training samples are equally import for ReID tasks. The existing works often prune out easy samples to speed up training procedure. Instead of focusing on the easy samples, this work proposed a comprehensive approach towards data pruning by taking into consideration the samples importance, correcting mislabels and outlier removal. Extensive results suggest the effectiveness with a large percentage of data pruning on ReID tasks.

**Audience:**

Yes

**Broader Impact Concerns:**

There is no ethical concerns.

**Claims And Evidence:**

No

**Requested Changes:**

- Additional comparisons with state-of-the-art methods are missing. In particular, comparisons with learning from noisy labeled data and outlier detection methods [1-3] are necessary.

- Additional discussions of learning from noisy labeled data [4-5] is also necessary as the core contribution lies on learning from noisy labeled data and outlier detection.

[1] Han B, Yao Q, Yu X, et al. Co-teaching: Robust training of deep neural networks with extremely noisy labels[J]. Advances in neural information processing systems, 2018, 31.
[2] Ma X, Huang H, Wang Y, et al. Normalized loss functions for deep learning with noisy labels[C]//International conference on machine learning. PMLR, 2020: 6543-6553.
[3] Li S, Xia X, Ge S, et al. Selective-supervised contrastive learning with noisy labels[C]//Proceedings of the IEEE/CVF Conference on Computer Vision and Pattern Recognition. 2022: 316-325.
[4] Song H, Kim M, Park D, et al. Learning from noisy labels with deep neural networks: A survey[J]. IEEE Transactions on Neural Networks and Learning Systems, 2022.
[5] Wang H, Bah M J, Hammad M. Progress in outlier detection techniques: A survey[J]. Ieee Access, 2019, 7: 107964-108000.

**Strengths And Weaknesses:**

Strength:

- Dataset quality is a realistic concern for all learning based approaches. Methods tackling data pruning has many downstream applications.

- Compared with previous works, this work jointly considers multiple factors in data pruning, including removing easy samples, correcting mislabels and eliminating outliers.

Weakenss:

- The proposed method is too naive. Correcting mislabels is simply achieved by resorting the pseudo label with the highest probability. This naive way of self-training is subject to confirmation bias [1] where the performance of network could be severely harmed by incorrect pseudo labels. Moreover, the pseudo label correction requires very careful calibration of model training procedure. Introducing the label correction too early may fail to fit the hard samples while too late could overfit to the label noise.

- The outlier detection approach relies on a predefined threshold. This also introduces additional hyper-parameter to tune and there is no principled way introduced for this hyper-parameter tuning.

- The description of pruning easy samples is not clear. A discussion of existing work AUM is brought in but the methodology section is expected to be self-contained. It is very hard to fully understand how this step is implemented in the current form.

- Comparisons with state-of-the-art methods are missing. In particular, comparisons with learning from noisy labeled data and outlier detection methods are missing, which are the core contributions of this work.

[1] Arazo E, Ortego D, Albert P, et al. Pseudo-labeling and confirmation bias in deep semi-supervised learning[C]//2020 International Joint Conference on Neural Networks (IJCNN). IEEE, 2020: 1-8.

---

> ### Author Response · Authors · 2024-03-11
>
> **First and foremost**, we sincerely appreciate the reviewer's detailed analysis of our paper. We understand the strengths and weaknesses pointed out. We will address the concerns raised.
> ## Responses to Weaknesses:
>
> ### (1a) Simplicity and naivety of our method
> **Response:** Our method is primarily designed for the task of data pruning, not label correction. In the primary task of data pruning, our approach significantly outperforms existing SOTA at a much lower computational cost as well.
> Based on our data pruning method, we can also conceive a label correction method that is intuitive and efficient because it does not need to train an additional and more complex network for label correction. Hence, it can be seamlessly integrated into our data pruning approach without any additional computational overhead. We will clarify this in the revised version.
>
> ### (1b) Reliability of pseudo labels for label correction
> **Response:** We acknowledge the phenomenon mentioned by the reviewer and observed it in Fig. 15 (Appendix). However, thanks to the idea of avg logit, our method demonstrates relative robustness to the number of training epochs. At a low noise rate (10%), after label correction the model's accuracy stabilizes after 8 epochs, showing minimal impact from the number of training epochs.
> Even at a higher noise rate (30%), accuracy can still reach consistently high levels between the 8th and 20th training epochs.
> In summary, our method demonstrates considerable robustness, especially at low noise levels. Even at higher noise rates, our approach is not overly sensitive to the number of training epochs, achieving good results across a wide range of training epochs.
>
> ### (2) Threshold for outlier detection
> **Response:** We present the impact of the outlier removal threshold in Fig.12. We observe that this threshold has a very limited impact on the final performance, the differences caused by different thresholds are almost negligible. Due to the negligible impact, developing a specific method for fine-tuning this threshold could become unnecessary.
>
> If we have to provide a fine-tuning method, we can refer to the AUM: Firstly, we manually create some outliers, e.g., simulating multi-target coexistence by merging two images or introducing heavy occlusion through extensive random erasing.
> Subsequently, these manually created outliers are included in the training dataset, and train a model on it once.
> Finally, the max class scores of these samples are recorded and can be used as the threshold.
> However, irrespective of the fine-tuning method employed, it inevitably increases computational overhead, deviating from the initial purpose of data pruning.
>
> ### (3) Writing in the methodology section
> **Response:** We appreciate the reviewer for the valuable suggestion. We will make it more self-contained and write a description of AUM in the next version.
> ### (4) Comparisons with state-of-the-art label correction and outlier detection methods
> **Response:** Our method is primarily designed for data pruning and additionally supports noisy sample removal, which existing SOTA data pruning methods do not. Accordingly, we benchmark our methods against other data pruning methods. Besides, in adherence to the primary objective of data pruning—to enhance training efficiency without incurring additional computational costs—we introduce a label correction algorithm, which is seamlessly integrated with our data pruning strategy, requiring no extra computational cost.
> We acknowledge that there are some more complex SOTA label correction methods, which may provide better results in learning from noisy data. However, these algorithms either require a more complex network structure [1] or training a model with another loss function [2] or employing a new model for selecting confident samples [3]. All of them inevitably increase the computational costs significantly, which deviates from our primary goal – minimizing computational overhead using data pruning. We therefore benchmark our method against the AUM method, which also avoids extra computational costs for outlier detection, ensuring a fair comparison under identical computational overhead. In summary, our approach not only adheres to the principle of computational efficiency but also demonstrates good performance in learning from noisy samples without additional computational costs.
>
> **References:**
>
> [1] Han B, Yao Q, Yu X, et al. Co-teaching: Robust training of deep neural networks with extremely noisy labels[J]. Advances in neural information processing systems, 2018, 31.
>
> [2] Ma X, Huang H, Wang Y, et al. Normalized loss functions for deep learning with noisy labels[C]//International conference on machine learning. PMLR, 2020: 6543-6553.
>
> [3] Li S, Xia X, Ge S, et al. Selective-supervised contrastive learning with noisy labels[C]//Proceedings of the IEEE/CVF Conference on Computer Vision and Pattern Recognition. 2022: 316-325.

---

### Decision · Action_Editor_jJCA · 2024-04-18

**Recommendation:** Accept with minor revision

**Comment:**

**Strengths:**
1. **Realistic Problem Addressing:** The paper addresses the practical and significant issue of data pruning in the RID (Re-Identification) task, highlighting its applicability in real-world scenarios.
2. **Promising Experimental Results:** The experiments conducted demonstrate promising results across three benchmark datasets within the proposed model settings, showing the potential effectiveness of the methodology.
3. **Comprehensive Experimentation:** The paper presents a thorough experimental approach that tests the method's efficacy on multiple datasets, providing robust evidence of its superiority in specific contexts.
4. **Innovative Metric Use:** The utilization of soft labels as average logits and entropy as an importance score for labels is noted as simple, intuitive, and effective, contributing to the method's overall strength.

The method is simple, and yet effect. Two of reviewers vote the acceptance. Particularly, one concerned weakness is
1. **Weak Connection to RID Task:** The proposed data pruning strategy's connection to the specific requirements of the RID task is weak, as it is largely based on a generic classification paradigm, making its specialized effectiveness for RID unclear.
2. **Typos and Minor Errors:** The paper contains various grammatical mistakes and typographical errors that could detract from its professional quality and clarity.
3. **Ambiguous Method Descriptions:** Descriptions of certain methodologies, such as pruning easy samples, are unclear and not self-contained, making it difficult for readers to grasp the implementation fully.

Please incorporate the rebuttal to improve the paper. Particularly, add some discussion in introduction or related work for Weakness 1 listed above. Overall, while the paper offers promising approaches and solid experimentation, it has incremental contributions that rely heavily on previous work.

**Audience:**

This paper is interested to the researchers in computer vision and machine learning communities.

**Claims And Evidence:**

This paper focuses on improving how data is pruned, which means removing unhelpful or irrelevant data points from a dataset. The authors review existing data pruning techniques and note that these techniques struggle with noisy (incorrect or misleading) data and fail to use all the available information during training. To address these issues, they introduce a new method that tracks the changes in pseudo labels (temporary labels assigned during training) over time and prunes data based on how consistent these labels are, measured by the entropy of the labels. Additionally, they develop strategies using this method to fix incorrectly labeled data and to spot outliers. The effectiveness of their approach is demonstrated through detailed experiments in both object re-identification and image classification tasks.

---

> ### Author Response · Authors · 2024-04-30
>
> Dear Action Editor,
>
> We would like to thank you for handling our paper and making a positive decision.
> Based on your revision suggestions and the points raised in our rebuttal, we have made the following revisions in the camera-ready version of our paper:
>
> 1. **Connection to RID Tasks**.
> We have refined the introduction section to better highlight the connection between our method and ReID task according to the following key ideas:
>
>     Our method is motivated by two issues of ReID datasets, i.e., less informative samples and dataset noise. To address the first issue, we explore whether (general) data pruning methods designed for classification tasks can be applied to ReID tasks. Next, we introduce a widely applicable data pruning approach to resolve both issues. The results of our experiments indicate that 30% of samples from the MSMT17 dataset can be pruned with negligible loss in accuracy. Our work is dedicated to closing the gap in applying (general) data pruning to ReID datasets.
>
>
> 2. **Typos and Minor Errors**. We have undertaken a comprehensive review to ensure that all of these issues are corrected.
>
>
> 3. **Method Descriptions**. In Sec. 3 and Sec. 5.5, we added more detailed descriptions to make our paper more self-contained and clearer, making it easier for readers to understand our methods fully.
>
> Once again, thank you for your invaluable support throughout the review process.
>
> Best regards,
>
> Authors